# Afadin couples RAS GTPases to the polarity rheostat Scribble

Marilyn Goudreault[1], Valérie Gagné[1], Chang Hwa Jo [1], Swati Singh[1], Ryan C. Killoran[1], Anne-Claude Gingras [2,3] & Matthew J. Smith [1,4] ✉

AFDN/Afadin is required for establishment and maintenance of cell-cell contacts and is a unique effector of RAS GTPases. The biological consequences of RAS complex with AFDN are unknown. We used proximity-based proteomics to generate an interaction map for two isoforms of AFDN, identifying the polarity protein SCRIB/Scribble as the top hit. We reveal that the first PDZ domain of SCRIB and the AFDN FHA domain mediate a direct but non-canonical interaction between these important adhesion and polarity proteins. Further, the dual RA domains of AFDN have broad specificity for RAS and RAP GTPases, and KRAS co-localizes with AFDN and promotes AFDN-SCRIB complex formation. Knockout of *AFDN* or *SCRIB* in epithelial cells disrupts MAPK and PI3K activation kinetics and inhibits motility in a growth factor-dependent manner. These data have important implications for understanding why cells with activated RAS have reduced cell contacts and polarity defects and implicate AFDN as a genuine RAS effector.

RAS small GTPases function as molecular switches by undergoing nucleotide-dependent conformational exchange. When GTP-bound, RAS GTPases are bound directly by effector proteins that are responsible for transmitting signals to diverse cellular pathways[1–4]. This is the accepted paradigm of GTPase signalling, yet most of our understanding derives from study of a small subset of GTPase-effector partners and the biological significance of many proposed RAS effectors remains uncertain (Fig. 1a).

The RAS subfamily of small GTPases consists of 35 members and includes the archetypal oncoproteins H-, K- and N-RAS, along with related GTPases of the RAP, RAL, RIT and RHEB families[5]. Effectors bind activated RAS GTPases through RAS Association domains (RAs) or RAS Binding Domains (RBDs). Structural biology studies have revealed that RBD/RA domains share a common ubiquitin fold and interact with the same epitope on RAS[6–9]. There are over 50 predicted RBD/RA domains in the human proteome present in an array of proteins with diverse associated domains[10]. While the RAF and PI3K effectors have received the most attention due to their role in proliferation and survival, there are numerous effectors that are highly conserved through evolution that bind activated RAS in vitro and in vivo. These include the RALGEF

proteins[11,12], the RASSF Hippo pathway-effectors[13–16] and the adhesion protein AFDN (also Afadin/AF6/MLLT4).

AFDN, an ortholog of *Drosophila* canoe, is a unique RAS effector with two N-terminal RA domains that plays an essential role in the formation and maintenance of adherens junctions (AJs)[17–19]. Originally identified as a fusion partner of the MLL histone methyltransferase in leukemia[20], AFDN was later shown to directly bind activated HRAS, RAP1A and RAP2A[1,21]. Further evidence links AFDN to RRAS and MRAS[22], revealing potentially plastic GTPase-effector signalling. In *Drosophila*, canoe regulates the connection between AJs and actin and maintains polarity during apical constriction in a RAP-dependent manner[18,19]. A crystal structure of the first RA domain (RA1) of AFDN bound to HRAS validated its capacity to associate with RAS via a ubiquitin-like domain[10], but a biological function for this interaction has not been explored. Intriguingly, recent application of proximity-based proteomics (BioID) has consistently found the most proximal effector of activated RAS in cells is not RAF or PI3K, but AFDN[23,24]. While RAS binding to AFDN may not directly stimulate proliferation, the metastatic nature of RAS mutant cancers makes study of this GTPase-effector complex a high priority.

[1]Institute for Research in Immunology and Cancer, Université de Montréal, Montréal, QC H3T 1J4, Canada. [2]Lunenfeld-Tanenbaum Research Institute, Mount Sinai Hospital, Toronto, ON M5G 1X5, Canada. [3]Department of Molecular Genetics, University of Toronto, Toronto, ON M5G 1X5, Canada. [4]Department of Pathology and Cell Biology, Faculty of Medicine, Université de Montréal, Montréal, QC H3T 1J4, Canada. ✉e-mail: matthew.james.smith@umontreal.ca

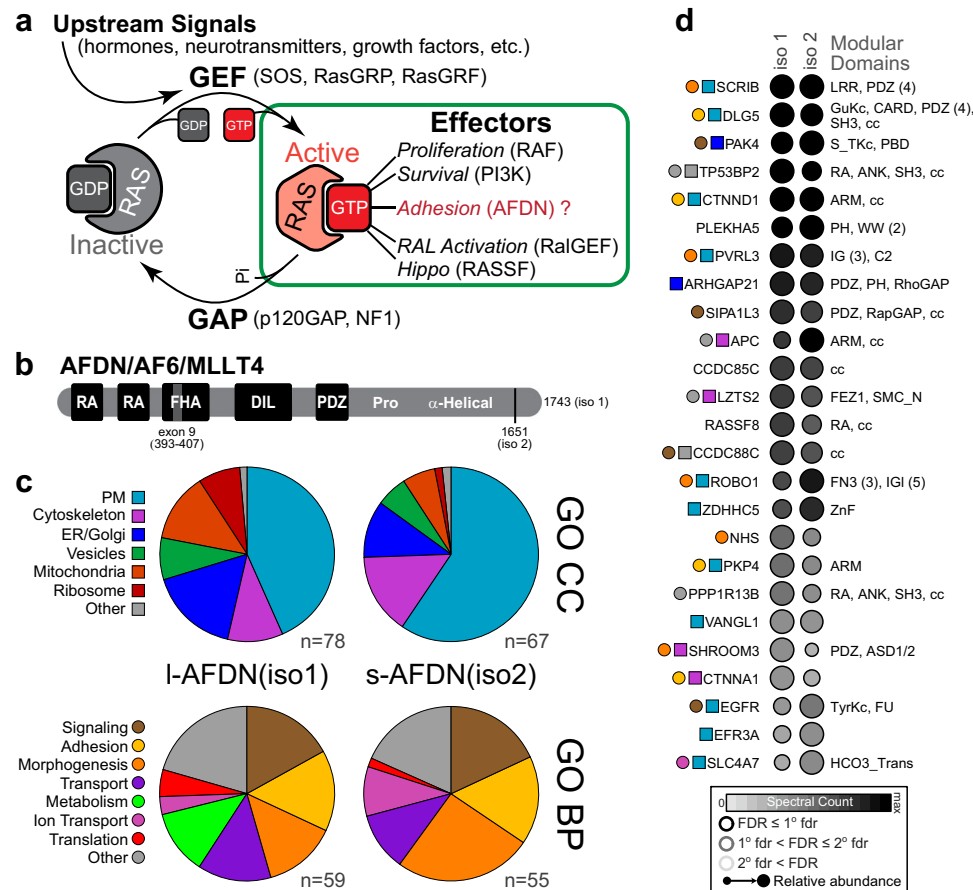

**Fig. 1 | BioID identification of proteins with in vivo proximity to AFDN.**
**a** Activation of RAS subfamily small GTPases is governed by differential binding to guanine nucleotides. In the GTP-bound state RAS interacts with effector proteins, including AFDN. **b** Domain organization of the AFDN protein. RA = RAS Association domain; FHA = forkhead-associated domain; DIL = dilute domain; PDZ = PSD95/ DLG1/ZO-1 domain. These are followed by a proline rich region and a C-terminal helical domain involved in actin binding. **c** High-confidence BioID hits (FDR ≤ 0.01) for both isoforms of AFDN were categorized by Gene Ontology (GO) Cellular Component (CC; top) or GO Biological Process (BP; bottom). **d** Top 25 BioID preys with their associated domains. Available GO CC (square) or GO BP (circle) terms are colored as in **c**. The relative abundance, FDR and spectral count are represented by shaded circles and denoted as per the legend (bottom).

Signal transduction from activated GTPases through effector proteins is dependent on their uniquely associated protein domains. For AFDN (Fig. 1b), this includes a forkhead-associated (FHA) domain with no known binding partners, a dilute (DIL) domain that binds ADIP to influence actin cytoskeletal organization at AJs[25], and a PSD95/DLG1/ ZO-1 (PDZ) domain with numerous binding partners involved in the cell adhesion apparatus (Nectins[26], JAM-A[27], and Eph receptors[28]). Other partners include the tight junction protein ZO-1[29], the AJ cytoskeletal regulator PLEKHA7[30] and the AJ components p120-catenin and α-catenin[29-31]. The C-terminal helical region of AFDN binds to actin, deemed critical to its role in regulating cell adhesion. This region is absent from a truncated isoform called s-AFDN (iso2) compared with the longer l-AFDN (iso1)[32,33]. A complete signalling network for AFDN has yet to be defined and the impact of GTPase binding on its interactors is unknown. Despite this, AFDN plays a key role in regulating cell adhesion in metazoans and signalling from RAS subfamily GTPases to AFDN could have important implications for metastasis in RAS-driven cancers. Indeed, AFDN mutations were identified as drivers in breast cancer[34], and loss of AFDN from cell-cell contacts induces migration and invasion in endometrial[35], pancreatic[36] and breast cancers[37] as well as glioblastomas[38].

Here, we use proximity labelling coupled to mass spectrometry to build a comprehensive protein interaction network for the RAS effector AFDN. Our data corroborate known partners and uncover many additional interactors, most notably direct binding to the tumour suppressor polarity protein SCRIB (ortholog of *Drosophila*

scribbled). We provide evidence that AFDN and SCRIB associate in cells via a non-canonical FHA-PDZ domain interaction, and that activated RAS GTPases promote formation of this complex. Loss of *AFDN* or *SCRIB* impacts RAS activation of the MAPK and PI3K pathways and disrupts motility in a growth factor-dependent manner. These results have important implications to both RAS-induced tumourigenesis and the normal development and maintenance of cell-cell contacts in epithelial layers.

## Results

### The AFDN interaction network

To understand the implications of AFDN as a GTPase effector we required a comprehensive map of the AFDN interaction network. To address this we used proximity biotinylation coupled to proteomics (BioID)[39]. For bait, we generated stable HeLa cell lines that express either the long (iso1) or short (iso2) isoforms of AFDN fused with biotin ligase (BirA*) and the FLAG epitope in a Tet-inducible manner (Supplementary Fig. 1a). Data from two biological replicates were filtered with SAINT[40], revealing a total of 95 high-confidence bait-prey relationships (Supplementary Data 1). Gene Ontology (GO) analysis of the full set of interactors provides insight to the localization and function of AFDN (Fig. 1c). Both AFDN iso1 and iso2 associate primarily with signalling, adhesion, transport and morphogenesis proteins at the plasma membrane, with evidence that either AFDN or its prey proteins shuttle to the endoplasmic reticulum (ER), Golgi and mitochondria. 46/95 (48%) of identified proteins were found using both AFDN iso1

and iso2 as baits, including the majority of those with the highest spectral counts. The top 25 coincident hits are listed in Fig. 1d. These proteins have a variety of associated domains but are particularly enriched in PDZ domains, typical of proteins involved in cell adhesion. As the majority of small GTPases in vivo are in the inactive, GDP-bound conformation we did not expect detection of transient GTPase partners using this approach, however, it is notable that several other RA domain proteins (and putative RAS effectors) are present in the dataset (TP53BP2/ASPP2, RASSF8, and PPP1R13B/ASPP1[16,41]). Several hits were isoform-specific and are listed in Supplementary Fig. 1b, c. These were primarily ER/Golgi proteins for iso1 of AFDN, and plasma membrane proteins for the shorter iso2 that is deficient in actin binding. There were several known AFDN partners identified that corroborate the efficacy of this approach, including cadherin-associated CTNND1/p120-catenin and CTNNA1/α-catenin, and the cell adhesion receptors PVRL2/nectin-2 and PVRL3/nectin-3. The most prominent preys identified using either bait, however, had not previously been associated with AFDN function: the master polarity protein Scribble (SCRIB) and the adhesion protein Discs Large Homolog 5 (DLG5). While association with all identified preys may have important implications for AFDN function, there are intriguing data linking SCRIB to RAS GTPase-induced metastasis[42–45] and regulation of AJs. Loss of *AFDN* and *SCRIB* synergistically promotes RAS-induced cell growth[46] and invasion[47], and canoe and scribbled are co-determinants of AJ formation in *Drosophila* that co-localize at apical contacts in epithelial layers[48]. We therefore chose to investigate a possible direct link between AFDN and SCRIB.

## Protein domain mapping the AFDN-SCRIB interaction

To corroborate the BioID result we first examined AFDN and SCRIB subcellular localization. In human epithelial HeLa cells, used for the proteomic analysis, we observed overexpressed AFDN and SCRIB co-localize around the cell cortex and in the cytoplasm (Fig. 2a and Supplementary Fig. 1d, e). As these proteins are typically studied in cells that establish AJs and apical-basal polarity, we immunostained endogenous AFDN and SCRIB in the mammary epithelial MCF7 cell line. Though SCRIB is generally considered a basolateral marker and AFDN apical, z-stack projections revealed substantial overlap in their position at cell-cell contacts (Fig. 2b). To test if these proteins form a complex, we co-expressed full-length SCRIB and AFDN with N-terminal epitope tags. Upon immunoprecipitation (IP) of AFDN we observed significant and specific co-precipitation of SCRIB (Fig. 2c). We could also detect endogenous SCRIB in an IP of endogenous AFDN (Fig. 2d). These data corroborate a direct AFDN-SCRIB complex.

As both proteins comprise several distinct domains, we sought to map the interaction to specific regions. First, bacterial expression constructs encoding N-terminal glutathione S-transferase (GST) fusions of defined fragments covering the length of AFDN were generated (Fig. 2e). Following their expression and purification from *E. coli*, the fragments were mixed with glutathione beads and lysates of mammalian cells expressing potential interaction partners. Figure 2f shows that both overexpressed SCRIB and endogenous SCRIB were precipitated specifically by a fragment encompassing the AFDN FHA domain. It also verified that the RA1-RA2 region binds activated KRAS4B (G12V, constitutive GTP-bound mutant and referred to hereon as KRAS) but not wild-type KRAS (predominantly GDP-bound in vivo). Mixing with lysates expressing either N- or C-terminal halves of SCRIB revealed an interaction with only the N-terminal region, comprising the leucine-rich repeats (LRRs) and the first of four PDZ domains (PDZ1). To resolve which region of SCRIB binds AFDN we generated GST-fusions of the SCRIB LRR and PDZ1 domains (Fig. 2g). These proteins were purified and mixed with lysates from cells expressing either AFDN iso1 or iso2. Figure 2h demonstrates that the PDZ1 domain of SCRIB, and not the LRR region, specifically precipitated iso1, iso2 and endogenous AFDN. These data suggested that the FHA domain of AFDN and the PDZ1 domain of SCRIB mediate an interaction between

the two proteins. To confirm this, we generated domain deletion constructs to express either AFDNΔFHA (missing residues 370-600) or SCRIBΔPDZ1 (missing residues 718-820) in context of the full-length proteins. Compared to wild-type, AFDNΔFHA did not co-precipitate with SCRIB (Fig. 2i) and SCRIBΔPDZ1 no longer bound AFDN (Fig. 2j). These data corroborate an FHA-PDZ1 mediated interaction and propose SCRIB as the first known binding partner of the AFDN FHA domain.

## A phospho-independent FHA-PDZ1 complex

We sought to determine the molecular mechanism of this interaction and whether these protein modules associate directly, aided by an available NMR structure of the AFDN FHA domain (PDBid 1WLN) and two structures of SCRIB PDZ1 (PDBids 5VWK[49] and 1X5Q). Archetypal PDZ domain binding motifs are found at the C-termini of proteins, though some PDZ interactions occur at atypical internal binding sites[50–52]. SCRIB PDZ1 precipitated both AFDN iso1 and iso2, suggesting the interaction does not involve the AFDN C-terminus. To exclude the possibility that our bacterially expressed GST-FHA domain was itself providing a PDZ binding motif, we generated variants with distinct C-termini. AFDN FHA domain constructs ending at residues 590 (-ENR), 599 (-PEL) or 610 (-RES) were equally efficient at precipitating SCRIB from cell lysates (Fig. 3a). This suggested the presence of a non-canonical, internal PDZ binding site in the AFDN FHA domain.

FHA domains are protein-interaction modules classically involved in binding to phosphothreonine (pTHR) motifs[53,54]. It was conceivable that a pTHR site exists in the SCRIB PDZ1 domain that serves as a binding site for AFDN. However, sequence and structural alignments revealed that the AFDN FHA domain does not contain two positively charged residues critical for coordinating pTHR interactions in other FHA domains (Fig. 3b and Supplementary Fig. 2a). To confirm this experimentally, we repeated the co-IP of full-length AFDN and SCRIB following incubation of the lysates with phosphatase (Fig. 3c). As predicted, treatment of the lysates with phosphatase did not disrupt AFDN-SCRIB complex. We also performed a complete phospho-proteomic analysis of SCRIB to identify potential phosphorylation sites in PDZ1 (Supplementary Fig. 2b). Several sites were mapped, but no SER, THR or TYR residues in the PDZ1 domain were found phosphorylated. We also tested whether the alternatively spliced exon 9 (ex9) of *AFDN*, which results in removal of residues 393-407 in a loop region of the FHA domain, would alter its complex with SCRIB. IP of AFDNΔex9 following its co-transfection with SCRIB demonstrated that ex9 residues do not affect SCRIB binding (Supplementary Fig. 3a, b). Together, these data indicate that the AFDN-SCRIB interaction is phospho-independent with no significant role for the alternatively spliced ex9, and their complex could be studied in vitro with purified proteins.

The SCRIB PDZ1 domain proved well folded and soluble using previously defined boundaries (residues 718-820). For the AFDN FHA domain, we generated five GST fusion constructs based on the available structure and on secondary structure predictions that revealed a helical region spanning residues 563-576 (Supplementary Fig. 3a). We mixed purified SCRIB PDZ1 domain with each of the recombinant FHA domains and found the predicted α-helix is essential for a direct interaction (Fig. 3d). The core FHA domain alone, as solved in the NMR structure (381-501), did not bind PDZ1. The importance of the extended C-terminal region was corroborated by NMR spectroscopy. [1]H/[15]N-HSQCs of [15]N-labelled AFDN FHA revealed the FHA domain fold is conserved in the extended protein, and chemical shifts specific to the C-terminal helix were identical to those demonstrating line broadening upon titration of unlabelled SCRIB PDZ1 (Fig. 3e and Supplementary Fig. 3c). Despite this, a construct encompassing only AFDN residues 531-580 did not bind SCRIB (Fig. 3f), indicating the helical extension and the FHA domain are both required for the PDZ1 interaction. This α-

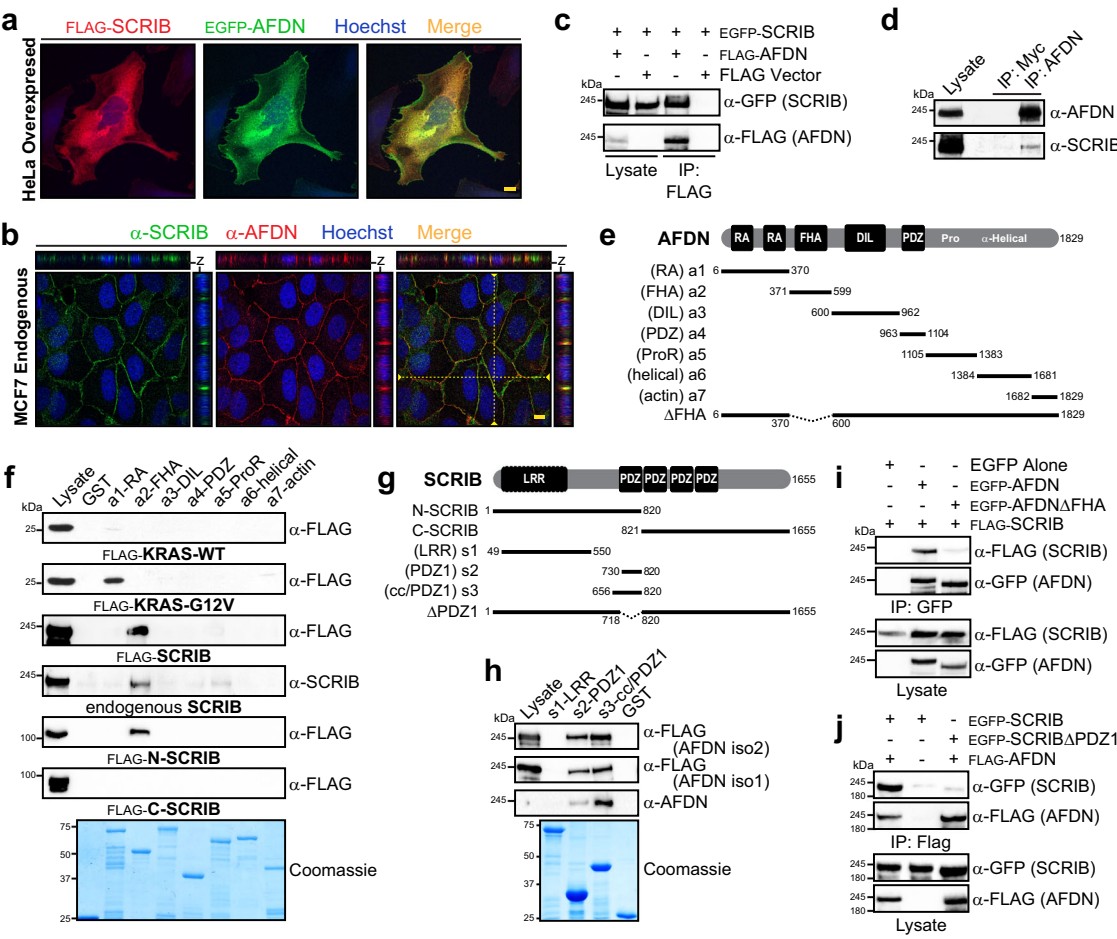

**Fig. 2 | A SCRIB-AFDN complex is mediated by the FHA domain of AFDN and the PDZ1 domain of SCRIB.** **a** Localization of AFDN and SCRIB in HeLa cells. EGFP-AFDN and immunostained FLAG-SCRIB were imaged by confocal microscopy. Scale bar represents 10 μm. **b** Endogenous SCRIB and AFDN co-localize at cell contacts in MCF7 cells. Antibody specificity to endogenous AFDN/SCRIB is verified in Fig. 6a. Projections of z-stacks are at right and top of the merged images, with position shown by dashed yellow line. Scale bar represents 10 μm. **c** Full-length AFDN and SCRIB co-immunoprecipitate (IP). EGFP-tagged SCRIB and FLAG-tagged AFDN were co-expressed in HEK 293T cells. IP with anti-FLAG followed by Western blotting with anti-GFP revealed the complex. Vector alone was a control. **d** Endogenous AFDN and SCRIB co-precipitate from MDCKII cells. Following AFDN IP, an anti-SCRIB blot confirmed the complex. No AFDN or SCRIB were observed in an anti-Myc IP (control). **e** Seven fragments comprising the whole of AFDN were generated to map the interaction, generally encompassing the AFDN modular domains. **f** SCRIB is

precipitated by the FHA domain of AFDN. GST-fusions of the 7 AFDN fragments were mixed with lysates expressing KRAS, activated KRAS-G12V, SCRIB, or the C/N-terminal regions of SCRIB. Full-length recombinant and endogenous SCRIB and its N-terminal half associated with AFDN FHA. Activated KRAS bound the RA domains. **g** Fragments of SCRIB used for mapping its interaction with AFDN. **h** AFDN was precipitated by the PDZ1 domain of SCRIB. GST-fusions of the SCRIB LRR or PDZ1 domains were mixed with lysates expressing AFDN. Both isoforms and endogenous AFDN were bound by PDZ1. **i** AFDN-SCRIB complex is dependent on the FHA domain of AFDN. Full-length AFDN or an FHA deletion mutant were co-transfected with SCRIB. Following anti-GFP IP, an anti-FLAG blot revealed SCRIB association is disrupted by AFDNΔFHA. **j** Complex between AFDN and SCRIB is dependent on the PDZ1 of SCRIB. Full-length or a PDZ1 deletion mutant of SCRIB were co-transfected with AFDN. Following anti-GFP IP, an anti-FLAG blot revealed SCRIBΔPDZ1 does not precipitate AFDN. All source data are provided in the Source Data files.

helix bears significant homology to a C-terminal helix in the Rad53 FHA domain[55], which stabilizes the FHA fold. Thus, we resolved a direct binding module comprising SCRIB PDZ1 (718-820) and a C-terminally extended FHA domain of AFDN (371-580).

To measure the affinity of purified FHA and PDZ1 domains we used isothermal titration calorimetry (ITC). The protein domains bound with an equilibrium dissociation constant ($K_d$) of 14.8 μM in 150 mM NaCl (Fig. 3g). This affinity is comparable to previously studied PDZ domain interactions[49,52,56,57]. To examine specificity, we purified the PDZ2 and PDZ3-4 domains of SCRIB and assessed their binding to the AFDN FHA domain. ITC analyses indicated these domains do not bind AFDN, confirming that its FHA domain binds specifically to SCRIB PDZ1 (Supplementary Fig. 4a, b). We also assessed whether deletion of ex9 residues in the FHA domain would moderate affinity. Purified FHA domain Δex9 bound PDZ1 with a $K_d$ of 22.9 μM, slightly weaker than the FHA domain with ex9 (Supplementary Fig. 4c). Finally, prototypical

PDZ domain interactions are driven by insertion of a hydrophobic residue into a deep hydrophobic pocket of the PDZ domain. To explore if there are electrostatic contributions to the FHA-PDZ1 complex we performed ITC at a lower salt concentration (20 mM NaCl; Supplementary Fig. 4c, d) and observed significantly improved affinity ($K_d$ of 5.7 μM). To map the interaction site on the SCRIB PDZ1 domain we used NMR spectroscopy. BMRBid 11207 was used to assign 71% of the peaks in a $^1H/^{15}N$-HSQC of $^{15}N$-labelled SCRIB PDZ1 (Supplementary Fig. 5a). Titration of unlabelled AFDN FHA domain resulted in extensive broadening of the majority of PDZ1 peaks, and significant chemical shift perturbations for some (Supplementary Fig. 5b). Figure 3h reports intensity ratios for all assigned peaks (unbound PDZ1 versus FHA domain-bound) and was used to map the binding interface. Using the structure of SCRIB PDZ1 bound to a C-terminal peptide from β-PIX (PDBid 5VWK) and our NMR data, we resolved that the AFDN FHA domain binds SCRIB PDZ1 in the classic C-terminal motif pocket

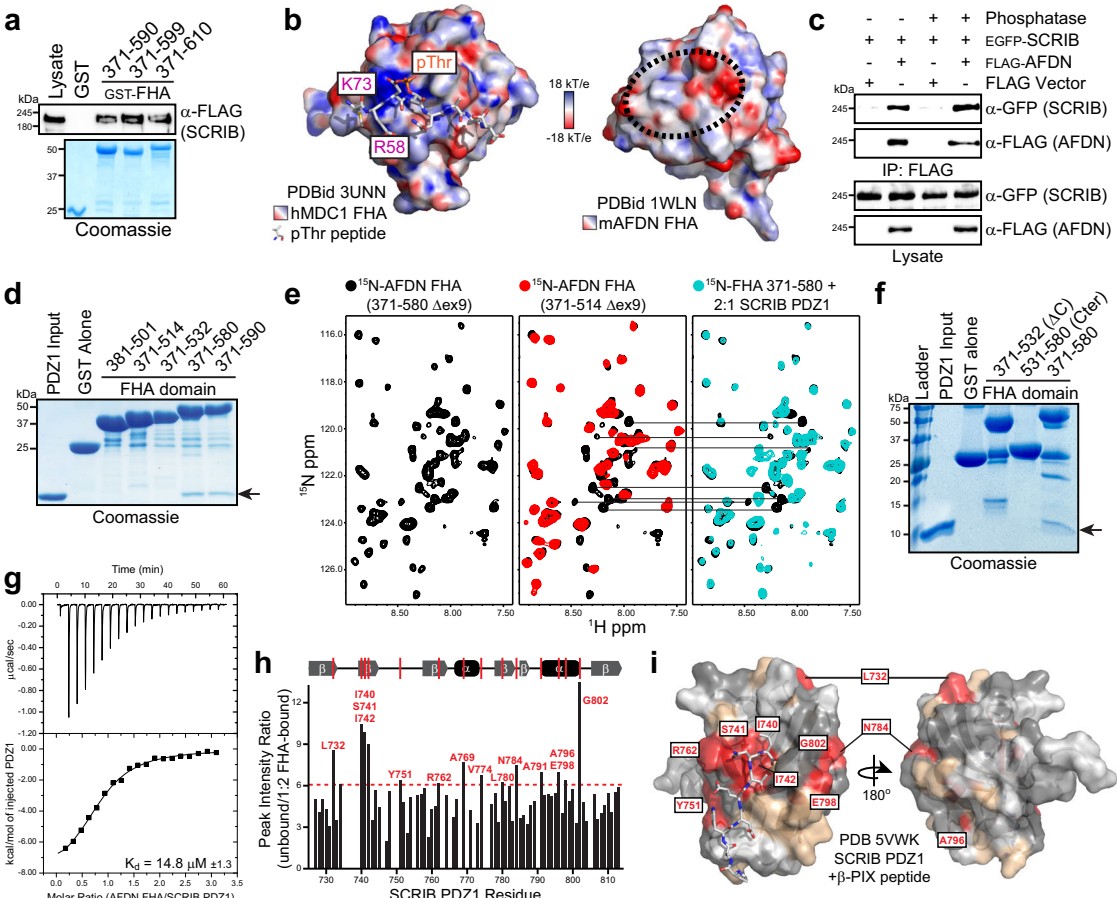

**Fig. 3 | AFDN FHA and SCRIB PDZ1 interact directly in a phospho-independent manner. a** GST-FHA domains with distinct C-termini precipitate full-length SCRIB from lysate. **b** Lack of positively charged residues on the surface of the AFDN FHA domain. The electrostatic surface of MDC1 FHA domain (*left*) shows the position of K73 and R58, necessary for binding pThr (PDBid 3UNN). Electrostatic surface of the AFDN FHA domain (PDBid 1WLN) reveals no positive charges (*right*; dashed circle). Electrostatic potential was calculated using APBS and PyMol. **c** AFDN and SCRIB co-IP following incubation with phosphatase. EGFP-SCRIB and FLAG-AFDN were co-expressed and precipitated with anti-FLAG following treatment with phosphatase. **d** Purified SCRIB PDZ1 can interact directly with GST-tagged AFDN FHA. Fragments end at FHA domain residues 501-590 and could only bind SCRIB PDZ1 if extended past residue 580. GST alone was a control. **e** ¹H/¹⁵N-HSQC spectra of the AFDN FHA domain. The extended FHA domain (371-580, black) overlays with the core FHA domain (371-514, red) and exhibits several additional peaks. The same peaks are broadened upon addition of unlabeled SCRIB PDZ1 (right, blue), verifying

importance of the extended C-terminal region to SCRIB binding. **f** The extended C-terminal FHA region alone does not bind SCRIB. Mixing PDZ1 domain of SCRIB with GST-tagged FHA domain fragments revealed neither the core FHA domain (371-531) nor C-terminal extension (531-580) in isolation complex with PDZ1. **g** ITC established AFDN FHA binds SCRIB PDZ1 with a $K_d$ of 14.8 μM, comparable to most PDZ domain interactions. **h** Plot of peak intensity ratios from ¹H/¹⁵N-HSQC spectra of PDZ1 alone vs PDZ1 in the presence of 2-fold molar excess AFDN FHA domain. Widespread broadening was observed, most impacted are labelled by residue number (over 6X difference, dashed red line). **i** Surface representation of the amino acids in SCRIB PDZ1 involved in binding the AFDN FHA domain (PDBid 5VWK). Residues demonstrating the most broadening are coloured red, moderate broadening in orange and no broadening in grey. Unassigned residues are white. Position of the β-PIX peptide is characteristic of PDZ interactions with C-terminal motifs. All source data are provided in the Source Data files.

(Fig. 3i). We have therefore elucidated a non-canonical FHA-PDZ domain interaction that is phospho- and C-termini-independent involving both electrostatic interactions and the archetypal hydrophobic PDZ domain binding pocket.

**RAS GTPases complex with AFDN-SCRIB**

AFDN has been shown to bind several RAS subfamily GTPases and is commonly described as a RAS/RAP effector. The dual N-terminal RA domains of AFDN are a unique and evolutionarily conserved feature. To determine whether activated GTPases could complex with AFDN and SCRIB we first considered KRAS. IP of AFDN following its co-expression with an activated mutant of KRAS (G12V) established both KRAS and endogenous SCRIB were co-precipitated with AFDN (Fig. 4a). A comparable level of SCRIB was detected with AFDN in the absence of activated KRAS, suggesting a ternary KRAS-AFDN-SCRIB complex. Similar results were obtained with endogenous AFDN when KRAS-G12V was co-expressed with FLAG-SCRIB (Fig. 4b). We then screened

whether AFDN could interact with several related RAS and RAP family GTPases. We purified the dual AFDN RA1/2 domains (residues 6-370) as a GST-tagged protein, as well as a GST-RBD from the archetypal RAS effector BRAF. We expressed 8 distinct RAS subfamily GTPases with activating mutations (constitutively GTP-bound, based on RAS G12V) in mammalian cells and mixed lysates with either GST-AFDN RA1/2 or GST-BRAF RBD (Fig. 4c). BRAF specifically bound the H-, K- and N-RAS GTPases, while AFDN demonstrated a broader specificity. In particular, the RAP GTPases RAP1B and RAP2C bound AFDN with similar intensity to H-, K- and N-RAS, and RAP1A to a lesser extent. The distantly related DIRAS2 showed no interaction. Of note, the dual RA domains of AFDN precipitated activated RAS with similar efficiency to BRAF despite a lower affinity of the single RA1 domain[10]. To dissect the specificity of the individual AFDN domains we purified RA1 and RA2 alone and contrasted their capacity to bind the RAS and RAP GTPases with the tandem RA1/2 domains. The RA1 of AFDN showed a very similar binding profile to RA1/2, though the levels of GTPase precipitated were

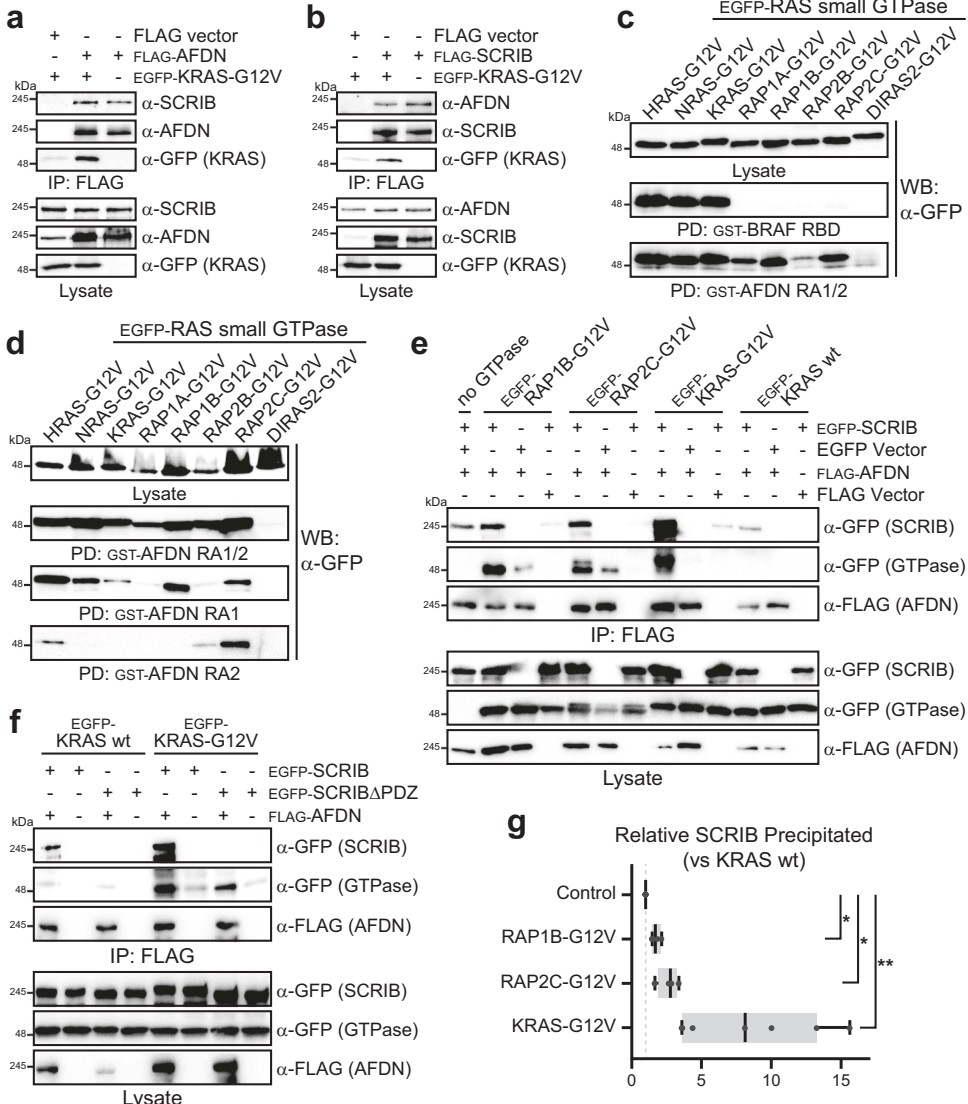

**Fig. 4 | RAS GTPases complex with AFDN-SCRIB. a** Activated KRAS is co-precipitated with AFDN and SCRIB. HEK 293T cells were co-transfected with vectors expressing FLAG-AFDN and EGFP-KRAS-G12V (constitutively active). Both endogenous SCRIB and KRAS-G12V were detected following anti-FLAG IP. FLAG vector alone was a control. **b** Activated KRAS and AFDN co-IP with SCRIB. HEK 293T cells were co-transfected with vectors expressing FLAG-SCRIB and EGFP-KRAS-G12V. Endogenous AFDN and KRAS-G12V were detected following anti-FLAG IP. **c** The RA1/2 domains of AFDN interact with multiple RAS small GTPases. EGFP-tagged GTPases with activating mutations were expressed in HEK 293T cells, and GST-tagged RBD domains from BRAF or AFDN purified from bacteria. GST-RBDs bound to glutathione beads were used to pull down (PD) GTPases, a Western blot with anti-GFP revealed interacting proteins. **d** Specificity of individual AFDN RA domains for RAS GTPases. GST-tagged RA1, RA2 or RA1/2 were purified and mixed with lysates expressing the indicated GTPases. An anti-GFP immunoblot following precipitation on glutathione beads revealed bound GTPases. Each blot was exposed for 30 seconds. **e** Activated GTPases co-precipitate with AFDN and augment interaction with

SCRIB. Full-length, EGFP-tagged SCRIB or activated RAS family GTPases were co-transfected with FLAG-tagged AFDN. Following immunoprecipitation with anti-FLAG, Western blot with anti-GFP revealed activated KRAS, RAP2C or RAP1B complex with AFDN and SCRIB. **f** SCRIB lacking PDZ1 does not co-precipitate with AFDN and does not prevent the association between AFDN and KRAS. Full-length wild-type or ΔPDZ1 SCRIB were co-transfected with AFDN and KRAS. Association with SCRIB and KRAS was detected by Western blot following anti-FLAG immunoprecipitation of AFDN. **g** Quantitation of SCRIB binding to AFDN when co-expressed with activated GTPases. The amount of EGFP-SCRIB co-precipitating with AFDN in anti-FLAG Western blots was determined by densitometry ($n = 4$ for RAP1B/RAP2C and $n = 7$ for KRAS, from distinct experiments). The ratio is the amount of SCRIB detected when co-expressed with RAP1B-G12V ($P = 0.018$), RAP2C-G12V ($P = 0.020$) or KRAS-G12V ($P = 0.007$), versus wild-type KRAS. Line represents the median, box the interquartile range (IQR) and whiskers the min/max. **$P < 0.01$, *$P < 0.05$ as measured by paired, two-tailed t-test. All source data are provided in the Source Data files.

considerably lower and this was particularly true of KRAS (Fig. 4d). The RA2 of AFDN had a more restricted specificity, precipitating only RAP2C with weaker binding to HRAS and RAP2B. Overall, the data reveal that AFDN interaction with RAP1B is driven by RA1, while interactions with other RAS GTPases are driven by avidity of the tandem domains. We further confirmed that full-length AFDN can associate with activated RAP1B, RAP2C and KRAS (Supplementary Fig. 6a). These data indicate that AFDN binds multiple GTPases in the

RAS subfamily and that SCRIB is maintained in an AFDN-KRAS complex.

How GTPases alter AFDN interactions is unknown. As AFDN and SCRIB may regulate adhesion and polarity downstream of RAS, we sought to examine how RAP or KRAS binding affects AFDN binding to SCRIB. We co-transfected full-length AFDN and SCRIB with KRAS-G12V, RAP1B-G12V or RAP2C-G12V and used wild-type KRAS or no GTPase as controls. We consistently observed increased SCRIB co-precipitating

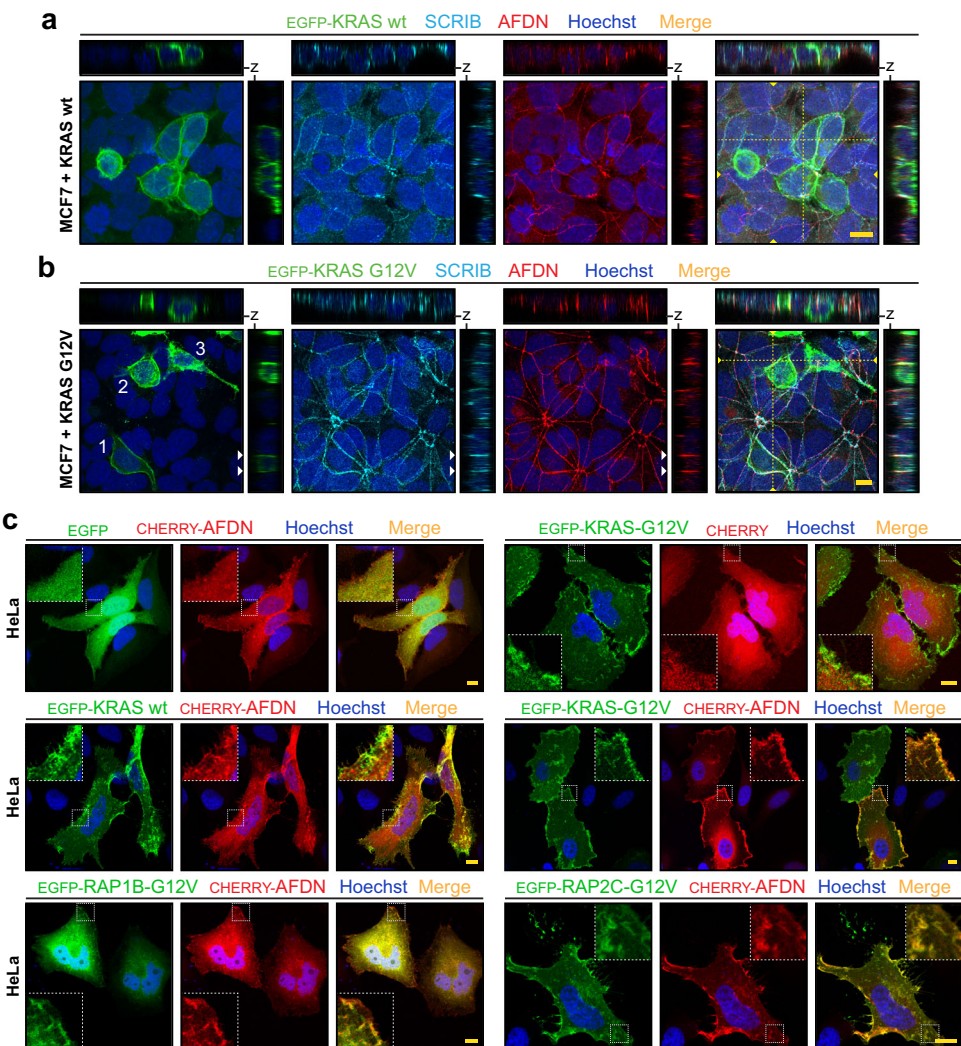

**Fig. 5 | RAS GTPases co-localize with AFDN. a** Expression of wild-type KRAS does not alter AFDN or SCRIB localization. EGFP-KRAS was expressed in MCF7 cells, and localization of endogenous AFDN and SCRIB was detected by immunostaining and confocal microscopy. Projections of z-stacks are at right and top. Position of associated z projections are shown with dashed yellow lines in the merged image. Scale bars represent 10 μm. **b** KRAS-G12V co-localizes with AFDN and SCRIB in MCF7 cells. Following expression of EGFP-KRAS-G12V, endogenous AFDN and SCRIB were detected by immunostaining. Cells expressing KRAS are found in the monolayer (1), detaching from the monolayer (2) or on top of the monolayer (3). Projections of z-stacks are at right and bottom of the merged images and their position is marked with a dashed yellow line. Scale bars represent 10 μm. **c** Activated RAS GTPases recruit AFDN to the cell membrane. Cherry-AFDN was co-expressed with EGFP-tagged RAS or RAP GTPases in HeLa cells. EGFP alone or EGFP-tagged wild-type KRAS with AFDN were controls, as was Cherry alone with activated KRAS-G12V. Scale bars represent 10 μm.

with AFDN in the presence of KRAS-G12V compared to RAP GTPases or controls (Fig. 4e). A SCRIB protein that can not bind AFDN (SCRIBΔPDZ1) did not alter AFDN association with KRAS-G12V and was not co-precipitated (Fig. 4f). Densitometry analysis of blots from distinct experiments indicated an 8-fold increase in SCRIB binding to AFDN when KRAS-G12V was co-expressed (compared to wild-type KRAS; Fig. 4g). RAP2C-G12V induced a more modest 2.6-fold increase in SCRIB precipitation, and RAP1B-G12V only a 1.7-fold increase. These data suggest that activated KRAS does not disrupt, and ostensibly augments, the AFDN-SCRIB complex.

We next considered whether AFDN, KRAS and SCRIB are co-localized in cells. All three proteins are typically located at the cortex of cells in epithelial layers, so we first chose to examine whether expression of oncogenic KRAS might disrupt the subcellular localization of AFDN and SCRIB in MCF7 cells (wild-type for *NRAS*, *KRAS* and *HRAS*). Expression of EGFP alone or EGFP-tagged, wild-type KRAS did not significantly alter AFDN or SCRIB localization and they did not co-localize with these proteins (Fig. 5a and Supplementary Fig. 6b, c).

Wild-type KRAS was observed ubiquitously across the plasma membrane, including along the basolateral surface. In contrast, transient expression of KRAS-G12V markedly disrupted cell-cell contacts with most detaching from adjacent cells within 24 hours (Supplementary Fig. 6d). At 48 hours, these cells were typically observed on the apical surface or underneath the monolayer. In the rare cells that remained intact within the monolayer, KRAS-G12V was noticeably co-localized with endogenous AFDN and SCRIB as determined by z-plane projections (Fig. 5b). Nonetheless, the constitutive membrane association of both KRAS and AFDN/SCRIB in polarized epithelial cells and the propensity for active KRAS to induce detachment made analyses of their co-localization challenging. To appropriately determine whether AFDN and KRAS can co-localize in cells, and whether this depends on GTP-loading, we therefore considered HeLa cells. This epithelial line expresses very low levels of endogenous AFDN, not detectable by immunofluorescence, and exogenous AFDN displays a cytoplasmic localization with minor enrichment at the membrane (Fig. 2a and Supplementary Fig. 1d, e). When KRAS-G12V was co-expressed with

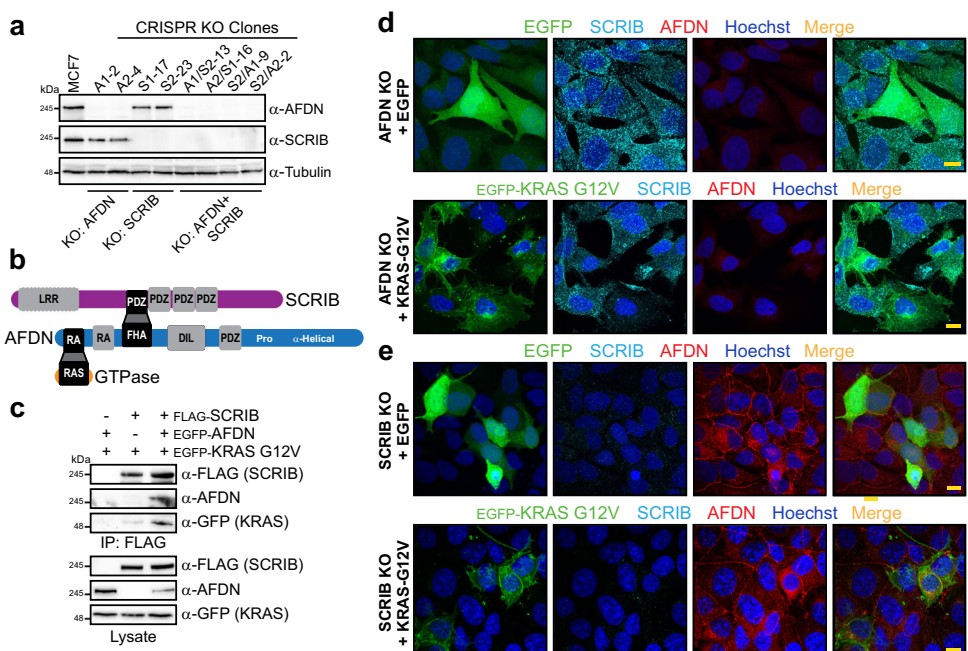

**Fig. 6 | KO of *AFDN* and *SCRIB* in MCF7 cells. a** Individual clones of MCF7 cells were selected following CRISPR KO of *AFDN*, *SCRIB* or both *AFDN/SCRIB* with two different guide sequences. Western blot with antibodies recognizing endogenous AFDN or SCRIB confirmed the proteins are not expressed. Source data are provided in the Source Data files. **b** Model of the SCRIB-AFDN-RAS GTPase complex based on our data. **c** KRAS does not complex with SCRIB in the absence of AFDN. FLAG-SCRIB was expressed in *AFDN* KO MCF7 cells together with EGFP-KRAS-G12V. Following IP with anti-FLAG, activated KRAS did not co-precipitate with SCRIB unless cells were also expressing EGFP-AFDN. Source data are provided in the Source Data files. **d** SCRIB is internalized and demonstrates a punctate pattern in *AFDN* KO MCF7 cells, which do not form cell-cell contacts (*top*). Expression of EGFP-KRAS-G12V in these cells does not alter SCRIB localization (*bottom*). Scale bars represent 10 µm. **e** AFDN remains predominantly at sites of cell contact in *SCRIB* KO MCF7 cells, which grow in multi-cell layers (*top*). EGFP-KRAS-G12V expression results in AFDN internalization and breakdown of cell-cell contacts (*bottom*). Scale bars represent 10 µm.

AFDN we observed extensive redistribution of AFDN to the membrane, completely co-localized with KRAS, with a small pool of AFDN retained in the perinuclear region (Fig. 5c). AFDN remained generally cytoplasmic in cells expressing wild-type KRAS, verifying their co-localization is dependent on RAS GTP-loading. We further examined whether activated RAP1B or RAP2C could similarly engage AFDN. Consistent with previous work, RAP1B-G12V was prominently distributed in the cytoplasm with a small pool at the plasma membrane that did co-localize with AFDN. Expression of RAP2C-G12V resulted in complete recruitment of AFDN to the cell membrane and endomembranes, with similar efficiency as KRAS. Thus, activated RAS GTPases are primed for AFDN complex and can recruit this effector to the cellular membrane.

## CRISPR KO of AFDN or SCRIB disrupts adhesion and polarity

To understand how loss of AFDN and/or SCRIB impacts RAS localization and further explore the KRAS-AFDN-SCRIB complex we generated CRISPR knockouts (KOs) of *AFDN*, *SCRIB* or both *AFDN* and *SCRIB* together in MCF7 cells (Fig. 6a). KO of *AFDN* produced a marked mesenchymal phenotype, with elongated and flattened cells lacking discernible cell-cell contacts, while *SCRIB* KO cells appeared to retain cell-cell adhesion but were smaller, rounder and grew in multicell layers (Supplementary Fig. 7a/b). We first sought to validate our elucidated model, whereby AFDN scaffolds a complex between SCRIB and RAS GTPases (Fig. 6b). This implies KRAS should no longer co-precipitate with SCRIB in cells lacking AFDN. To test this, we overexpressed activated KRAS-G12V and SCRIB in the presence or absence of AFDN in *AFDN* KO cells. Figure 6c shows KRAS only co-precipitated with SCRIB when AFDN was re-expressed, supporting the model. We next examined the localization of AFDN, SCRIB and KRAS in the KO cell lines. SCRIB localization was completely disrupted in *AFDN* KO cells, with endogenous SCRIB exhibiting a punctate pattern. We considered

whether these were early, late or recycling endosomes by expressing EGFP-tagged RAB5A, RAB7A or RAB11A, respectively (Supplementary Fig. 7c). SCRIB did not co-localize with these markers and the basis for the punctate distribution remains to be elucidated. Expression of KRAS-G12V in these cells did not alter SCRIB localization, with KRAS typically located to the plasma membrane (Fig. 6d). Wild-type KRAS displayed a similar localization (Supplementary Fig. 7d). In *SCRIB* KO cells, AFDN was retained at sites of cell-cell contact (Fig. 6e). z-plane projections of these regions could not distinguish whether the apical-basal distribution of AFDN is altered, as *SCRIB* KO cells do not form an organized monolayer but rather grow in layers 1-3 cells thick. We could not find *SCRIB* KO cells expressing KRAS-G12V that remained within the monolayer. Cells were instead observed on the apical or basal surfaces of cells not expressing KRAS. As with the parental MCF7 line, endogenous AFDN did not appear at cell contacts in KRAS-G12V expressing cells, while those expressing wild-type KRAS were retained in the monolayer and AFDN remained at cell contacts (Supplementary Fig. 7e). Together, these results corroborate a KRAS-AFDN-SCRIB complex and suggest that KRAS is membrane localized and presumably functional in cells lacking AFDN or SCRIB.

To further study the AFDN-SCRIB complex we generated rescue lines expressing either full length AFDN or AFDNΔFHA in *AFDN* KO cells, and full length SCRIB or SCRIBΔPDZ1 in *SCRIB* KO cells. Lentiviruses encoding these proteins with N-terminal EGFP tags were transduced (or EGFP alone control), and stable cell lines generated by selection (Supplementary Fig. 7f). We first examined whether re-expression of AFDN would restore SCRIB localization at the cell cortex of *AFDN* KO cells. Indeed, expression of either wild-type or ΔFHA AFDN induced SCRIB membrane localization and re-established cell-cell contacts, while SCRIB remained cytoplasmic in the EGFP alone control (Supplementary Fig. 8a). AFDN was retained at sites of cell contact in *SCRIB* KO cells, and this did not change with re-expression of SCRIB or

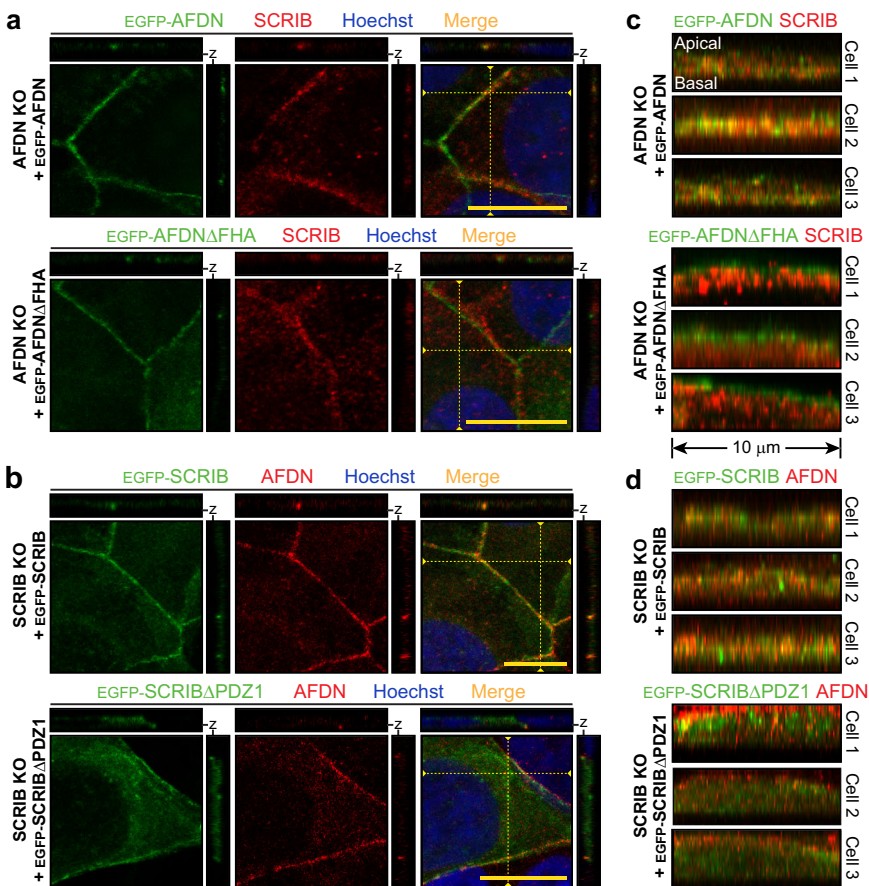

**Fig. 7 | Co-localization of SCRIB and AFDN is dependent on their PDZ1 and FHA domains, respectively. a** Expression of EGFP-AFDN in *AFDN* KO MCF7 cells restores endogenous SCRIB localization at cell contacts. SCRIB is also recruited to cell contacts in *AFDN* KO cells expressing EGFP-AFDNΔFHA but appears poorly co-localized with AFDN. Projections of *z*-stacks are at *right* and *top* and their position is marked with a dashed yellow line in the merged image. Scale bars represent 10 μm. **b** EGFP-SCRIB co-localizes with endogenous AFDN in *SCRIB* KO MCF7 cells. EGFP-SCRIBΔPDZ1 is also enriched at cell contacts in *SCRIB* KO cells. Projections of *z*-

stacks are at *right* and *top* and their position is marked with a dashed yellow line in the merged image. Scale bars represent 10 μm. **c** *z*-stack projections along a 10 μm length of cell contacts show exogenous, wild-type AFDN and endogenous SCRIB localization overlaps in *AFDN* KO cells (*top*, 3 independent cells). Conversely, expressed AFDNΔFHA remains apical to endogenous SCRIB (*bottom*). **d** *z*-stack projections show exogenous, wild-type SCRIB and endogenous AFDN co-localize in *SCRIB* KO cells (*top*) while SCRIBΔPDZ1 remains basal to endogenous AFDN (*bottom*).

SCRIBΔPDZ1 (Supplementary Fig. 8b). A closer examination of z-plane projections revealed that both AFDNΔFHA and SCRIBΔPDZ1, though enriched at the cell cortex, were also present more generally in the cytoplasm and did not completely overlap with endogenous SCRIB or AFDN, respectively (Fig. 7a, b). This could be directly observed when we considered projections of 10 μm length along cell-cell contacts, for AFDN rescues in Fig. 7c and for SCRIB rescues in Fig. 7d. These data show in multiple independent cells that wild-type AFDN and SCRIB localization is overlapping. Conversely, AFDNΔFHA remained apical to endogenous SCRIB in *AFDN* KO cells, and SCRIBΔPDZ1 basal to endogenous AFDN in *SCRIB* KO cells. Pearson coefficients verified a significant loss of co-localization between endogenous AFDN or SCRIB and the domain deletion rescues, as compared to wild-type (Supplementary Fig. 8c). This supports a model whereby a direct FHA-PDZ1 interaction is essential for proper localization of AFDN and SCRIB at cell contacts with apical-basal polarity.

### Loss of AFDN or SCRIB alters RAS signalling and cell motility in response to EGF

Finally, we explored whether loss of *AFDN* or *SCRIB* would alter signalling downstream of RAS. Stimulation of starved cells by growth factor (EGF) was used to promote activation of endogenous RAS. We first assessed whether the two major RAS proliferation and survival pathways, MAPK and PI3K, were activated and whether the kinetics of

this activation were altered. Time courses of EGF stimulation in the parental MCF7 line, the *AFDN* or *SCRIB* KO lines were performed, and pathway activation measured by induction of pERK (MAPK) or pAKT (PI3K) (Fig. 8a–c). Quantification of pERK (Fig. 8d) and pAKT (Fig. 8e) from three individual experiments revealed a clear defect in activation of both signalling pathways, with a sharp initial induction of pERK/pAKT but a rapid decrease in these phosphoproteins compared to the parental line. Interestingly, the defect was identical in both *AFDN* and *SCRIB* KO cells. This implies that an AFDN-SCRIB complex or their proper localization at cell-cell contacts could be important for regulating the RAS-RAF or RAS-PI3K signalosomes. We used a luminescence-based cell viability assay to determine if altered signalling kinetics would impact proliferation, but EGF alone was not able to induce proliferation of the parental or KO lines. When we measured proliferation in complete media (10% serum) there was a small but significant reduction in proliferation of *SCRIB* KO cells, while *AFDN* KO cells proliferated at the same rate as the parental line (Fig. 8f). As serum is not a specific RAS pathway activator, we can only conclude that loss of *AFDN*/*SCRIB* does not universally alter MCF7 proliferation.

Cancers driven by oncogenic RAS are characterized by predisposition to metastasis and malignancy. As AFDN and SCRIB are key regulators of adhesion and polarity we tested whether loss of *AFDN* or *SCRIB* would impact cell motility. In media containing serum we measured wound closure of MCF7, *AFDN* or *SCRIB* KO cells over a 72-

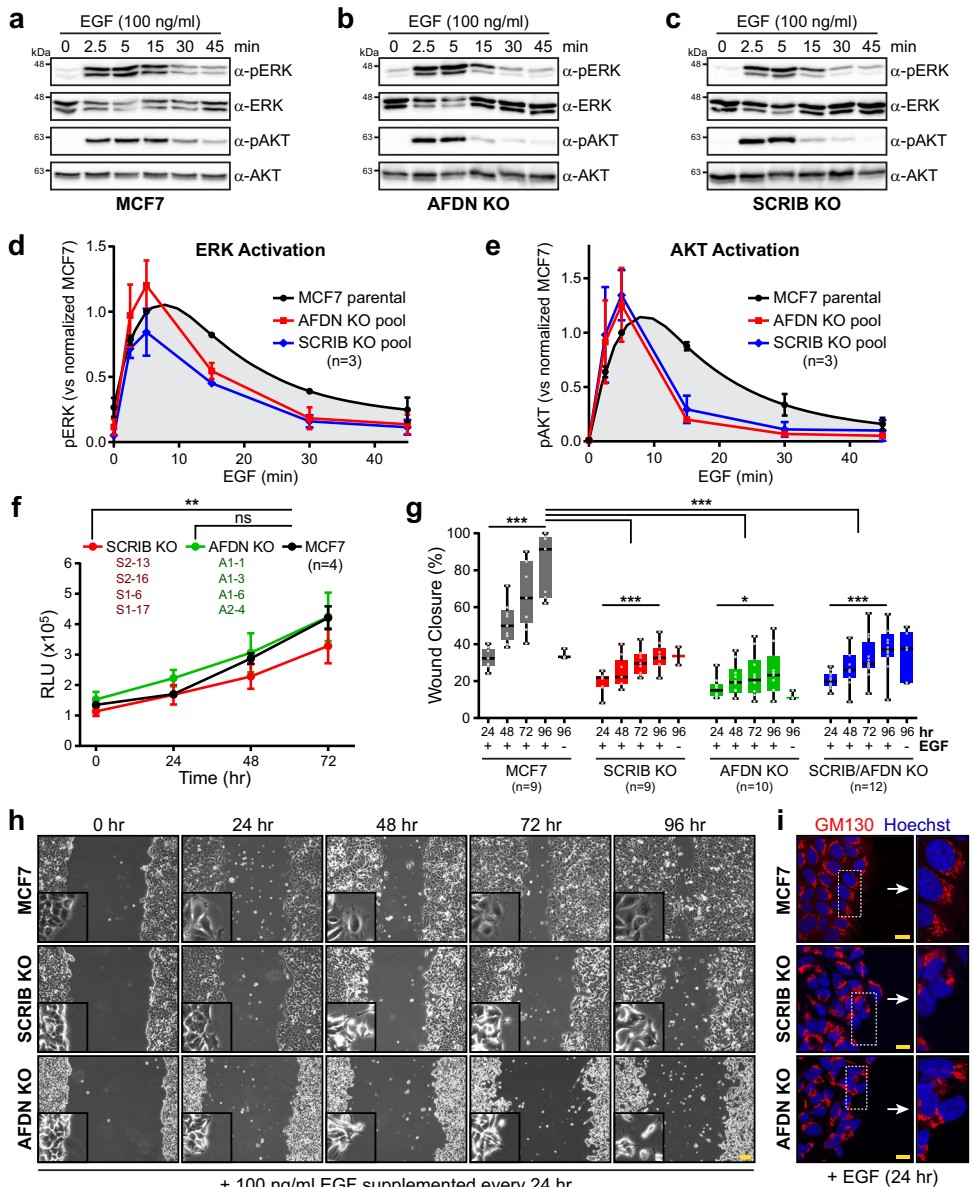

**Fig. 8 | Loss of *AFDN* or *SCRIB* disrupts ERK and AKT activation kinetics and cell motility in a growth factor-dependent manner.** Activation of the MAPK and PI3K pathways following EGF stimulation of MCF7 parental (**a**), *AFDN* KO (**b**) or *SCRIB* KO (**c**) cells. pERK or pAKT Western blots measure pathway activity 2.5, 5, 15, 30 or 45 minutes after addition of EGF. Total ERK and AKT Western blots confirmed expression and serve as loading controls. Quantification of MAPK (**d**; ERK activation) and PI3K (**e**; AKT activation) activity following addition of EGF over time. Center is the mean and error bars SD as derived from *n* = 3 independent replicates. **f** Effect of *AFDN* or *SCRIB* KO on cell proliferation compared to parental MCF7 cells over a 72-hour time course from *n* = 4 independent replicates. Center represents mean and error bars SD. **P < 0.005 (*P* = 0.003), ns = not significant (*P* = 0.298) as measured by two-way ANOVA. **g** KO of *SCRIB* or *AFDN* results in motility defects in response to EGF. Wound closure (%) was measured at 24, 48, 72 or 96 hours for MCF7 (*P* = 0.0001 within group), *AFDN* KO (*P* = 0.0145), *SCRIB* KO (*P* = 0.0001), and

*AFDN/SCRIB* KO (*P* = 0.0001) cells. 100 ng/ml EGF was supplemented to the media at time 0 and every 24-hour interval thereafter. *n* values denote independent replicates. Line represents the median, box the IQR and whiskers min/max. ***P < 0.001, *P < 0.01 as measured by RM one-way ANOVA (within group) or two-way ANOVA (between group). *P* = < 0.0001 for all between group comparisons. **h** Phase contrast images of MCF7, *SCRIB* KO and *AFDN* KO cells during wound closure. 100 ng/ml EGF was supplemented to the media every 24 hours. Inset are enlarged images of cells at the wound edge. Scale bar represents 100 μm. **i** *AFDN* and *SCRIB* KO cells do not polarize towards the wound. Cells were stained for GM130 (Golgi), which typically orients towards the wound at the leading edge (see parental MCF7 cells; *top*). Cells were fixed 24 hours after wounding and EGF stimulation. Arrows indicate direction of motility. Scale bars represent 20 μm. All source data are provided in the Source Data files.

hour time course (Supplementary Fig. 8d, e). All cell lines could migrate into and close the wound, with no significant difference between their motility rates. We then attempted the same experiment but using only growth factor (EGF) as a specific RAS-pathway activator (Fig. 8g, h). Stimulation of MCF7 cells with EGF every 24 hours for a total of 96 hours resulted in near complete wound closure. Conversely, KO lines of *AFDN*, *SCRIB* or *AFDN/SCRIB* demonstrated a complete loss

of motility. We hypothesized that loss of adhesion and polarity may account for this defect, and immunostained cells at the leading edge with anti-GM130, a Golgi marker. Golgi will characteristically orient toward the wound in migrating cells, and Fig. 8i shows this occurs in the MCF7 line. In contrast, both *SCRIB* and *AFDN* KO cells show randomly oriented Golgi that was not coordinated with wound direction. Thus, AFDN-SCRIB may be important mediators of adhesion and

polarity that regulate the ability of RAS GTPases to promote motility in response to growth factors.

## Discussion

Binding of RAS GTPases to multiple effectors with diverse signalling properties allows these molecular switches to act as gatekeepers. Outside a subset of well-studied effectors (RAF, PI3K), however, the biological consequences of RAS GTPase interactions with most RA/RBD domain proteins remains unknown. It is particularly intriguing that proteomics approaches consistently identify AFDN as the most proximal RAS effector[23,24]. Here, we have generated an AFDN interaction dataset, and the combination of previously known and novel interactors provides a network library through which an AFDN-RAS complex might function. A top hit was the adhesion/polarity tumour suppressor protein SCRIB. Several lines of evidence suggest that loss of SCRIB function plays a role in metastatic conversion of RAS transformed cells. In *Drosophila*, where deletion of *scribbled* has become a leading model to study RAS oncogenesis[46,58–62], neoplastic tumours arise from cells with both *scrib* and *Ras* mutations[44]. There is also substantial data linking SCRIB to regulation of the ERK/MAPK pathway in mammalian systems[63–66]. A direct molecular path linking SCRIB to RAS, however, has never been elucidated. We show that KRAS colocalizes with AFDN in cells and that this is dependent on RAS GTP-loading. Further, the AFDN-SCRIB complex is augmented by the presence of KRAS-G12V, an alteration of function that substantiates AFDN as a genuine RAS effector. A complete biological function for the RAS-induced stimulation of AFDN-SCRIB binding requires further work, but this effector module could play an important role in the maintenance and stability of cell-cell contacts and polarity in cells harbouring activated RAS. Conversely, in normal cells RAP GTPases may also function to position AFDN-SCRIB at emerging sites of cell-cell adhesion to initiate and couple AJ formation to establishment of polarity.

In addition to SCRIB there are several noteworthy hits in the AFDN BioID dataset, many of which are tumour suppressors. The presence of the RA domain proteins ASPP2[67], RASSF8[41] and ASPP1[16] implies a dynamic network of RAS subfamily effectors at the cell membrane. The specificity of these proteins for GTPases has not been well defined, but their presence suggests that multiple RA domain effectors and multiple RAS GTPase 'switches' could read diverse inputs at subcellular domains like AJs, and transduce signals to the appropriate pathways dependent on GTPase-effector specificity and effector-associated protein domains. RASSF8 and the ASPP proteins are tumour suppressors proposed to function in the Hippo and p53 pathways, respectively[68,69]. Other intriguing interactors include ROBO1, NOTCH2 and SLITRK4 that implicate AFDN in development and neurogenesis;[70–73] PAK4 and ARHGAP21 which regulate RAC1/CDC42 cytoskeletal remodelling;[74–76] DLG5, a membrane-associated guanylate kinase (MAGUK) protein involved in Hippo signalling;[77,78] and numerous mitochondrial, Golgi and ER proteins which suggest a role for trafficking from the plasma membrane to these intracellular organelles.

The AFDN network is highly enriched in proteins with PDZ domains. These prevalent domains typically bind with moderate affinity to the C-termini of multiple targets, scaffolding a dynamic network of cell adhesion, polarity and cytoskeletal proteins. PDZ domains have evolved numerous mechanisms to perform these scaffolding functions, including binding to internal peptide motifs[52,79]. Of the four PDZ domains in SCRIB, PDZ1 has considerably fewer binding partners than PDZ2-4 (β-PIX and *Drosophila* Gukh[49,80]). The enrichment of PDZ domains in our proteomic dataset suggests one or more may bind AFDN in a similar manner to SCRIB. It is also noteworthy that the AFDN FHA domain has no previously recognized binding partners. The lack of positively charged ARG or LYS residues in this domain is strong evidence that it does not bind pTHR motifs and has instead evolved a unique binding mode. Related KIF family FHA domains also appear to

lack phospho-dependency, and the kinesin KIF13 interacts with an adaptor CENTA1 independent of phosphorylation[81]. While we await elucidation of a structure for the AFDN-SCRIB complex, the non-canonical binding mechanisms highlight the complexity of studying protein-protein interactions based on predicted molecular functions.

Loss of function approaches are key to exploring the emerging connection between adhesion/polarity complexes and cellular transformation. To date, such experiments have provided a complex and inconsistent account of AFDN and SCRIB function. This includes numerous observations of both increased or decreased cellular proliferation, motility or invasion[37,82–84]. This is likely due to both context and cell type dependence as well as the use of RNA-mediated knockdown approaches which typically retain a significant level of protein expression. Here, we have generated CRISPR knockout clones of MCF7 epithelial cells that completely lack AFDN or SCRIB expression. We observed little difference in the ability of these cells to proliferate or collectively migrate in the presence of serum, but significant defects in response to growth factor. EGF is a potent activator of RAS, and loss of AFDN or SCRIB disrupted MAPK and PI3K induction as well as migration. Duration of ERK activation has long been recognized as a key component of cellular response to growth factor[85]. Sustained activation of ERK is necessary for G1 to S-phase transition and variation in ERK activation kinetics may influence proliferation versus differentiation outcomes[86,87]. The defect we observed suggests AFDN-SCRIB may compete with RAF and PI3K effectors and help regulate the duration of MAPK/PI3K signaling downstream of KRAS. RAP GTPases could further influence this by challenging KRAS for the AFDN RA domains, and indeed discrepancies in ERK signal duration downstream of some growth factors are RAP1-dependent[88]. Unfortunately, we were unable to examine AFDNΔFHA or SCRIBΔPDZ1 rescue lines in these experiments (or wild-type) as expression of these proteins was rapidly inactivated within 24-48 hours in culture (Supplementary Fig. 8f). This was observed independent of the tag or promoter used (CMV or EF-1α). Further, re-expression of AFDN in *AFDN* KO cells was not able to rescue E-cadherin localization to AJs (Supplementary Fig. 8g), indicating there has been a re-programming of these cells that cannot be reverted by simply re-expressing AFDN. This underscores the complex interplay of proteins at sites of cell-cell contact, whose positioning has a significant impact on gene expression and differentiation.

Future studies are required to elucidate comprehensive GTPase binding profiles for most proteins with RA or RBD domains, allowing systematic coupling of GTPases and effectors. Until then, it is essential to understand the biological processes impacted by specific effectors. Here, we've provided a network map for AFDN, a RAS GTPase effector that appeared early in the evolution of metazoans. The described interaction with the polarity protein SCRIB reveals a GTPase-AFDN-SCRIB axis that has implications for the formation and maintenance of cell adhesion during development, and the disruption of these processes during metastatic evolution in RAS-transformed cells.

## Methods

### Plasmid constructs and antibodies

For proteomic analysis we generated BirA*/FLAG-tagged mammalian expression constructs by Gateway cloning (Invitrogen) murine *Afadin* (GeneID 17356; residues 1-1820 for l-AFDN/iso1, or 1-1647 for s-AFDN/iso2) as entry vectors and then shuttling into BirA*/FLAG pcDNA5 FRT/TO. These were also placed in mammalian expression vectors with N-terminal FLAG or EGFP/Venus tags. AFDN fragments were constructed in the same manner, as were full-length and fragments of human SCRIB (GeneID 23513; cDNA a kind gift from Dr. Senthil Muthuswamy (Beth Israel Deaconess Medical Center at Harvard)). Fragments of AFDN and SCRIB for interaction mapping or biophysical analyses were constructed by sub-cloning the indicated regions into pGEX-4T2 (Amersham Pharmacia Biotech) with N-terminal glutathione *S*-transferase (GST) tag for bacterial expression. Gateway entry vectors

encoding mutationally activated human small GTPases KRAS4B (GeneID 3845), NRAS (GeneID 4893), HRAS (GeneID 3265), RAP1A (GeneID 5906), RAP1B (GeneID 5908), RAP2B (GeneID 5912), RAP2C (GeneID 57826), and DIRAS2 (GeneID 54769) were a kind gift from Dr. Jean-François Côté (IRCM, Montreal). Gateway entry vectors encoding human RAB5A (GeneID 5868), RAB7A (GeneID 7879) and RAB11A (GeneID 8766) were generated by gene synthesis. GTPases were shuttled into mammalian expression vectors with N-terminal EGFP or FLAG tags. Point mutations and deletions were performed by PCR-directed mutagenesis or Gibson assembly. All constructs were verified by sequencing. Primer sequences are in Supplementary Table 1. Antibodies used were: anti-AFDN polyclonal (Abcam, ab203569; WB: 1:500), anti-AFDN monoclonal (Abcam, ab90809/clone 3; WB: 1:200, IF: 1:100), anti-SCRIB (Cell Signalling, 4475 S; WB: 1:1000, IF: 1:50), anti-γ-Tubulin (Sigma, T6557/clone GTU-88; WB: 1:5000), anti-Myc (Roche, 11667149001), anti-GFP (Abcam, ab290; WB: 1:5000), anti-FLAG M2 (Sigma, F3165; WB: 1:1000, IF: 1:200), anti-E-Cadherin (BD, 610181/ clone 36; WB: 1:5000), anti-phospo-ERK1/2 (Cell Signalling, 9101; WB: 1:1000), anti-phospho-AKT (Cell Signalling, 4060; WB: 1:1000), anti-ERK1/2 (Cell Signalling, 4695; WB: 1:1000) anti-AKT (Cell signalling, 4691; WB: 1:1000), anti-mouse Tx-Red (Sigma, SAB3701076; WB: 1:100), anti-Rabbit Alexa Fluor 647 (Life Technologies, A-21244; WB: 1:200), anti-rabbit Cy5 (Abcam, ab6564; WB: 1:200), anti-rabbit Alexa Fluor 488 (Life Technologies, A-27034; WB: 1:200), HRP-conjugated anti-rabbit (Cedarlane, NA934; WB: 1:10000) and HRP-conjugated anti-mouse IgG (Fisher, 45-000-679; WB: 1:10000).

## BioID-Mass spectrometry

BioID[39,89] was done using HeLa Flp-In T-REx cells (Thermo Fisher, R71407) stably expressing BirA*/FLAG-tagged AFDN fusion proteins grown in $2 \times 150$ cm$^2$ plates of sub-confluent cells (60%) incubated 24 hours in complete media supplemented with 1 μg/ml tetracycline (BioShop) and 50 μM biotin (BioBasic). Cell pellets were resuspended by pipetting up and down and vortexing in 1.5 ml of RIPA buffer (50 mM Tris-HCl pH 7.5, 150 mM NaCl, 1% NP-40, 1 mM EDTA, 1 mM EGTA, 0.1% SDS, protease inhibitors (Sigma), and 0.5% sodium deoxycholate). 1 μl of benzonase (250U) was added to each sample and the lysates sonicated on ice. Lysates were centrifuged for 20 min at $12,000 \times g$, and then incubated with streptavidin-sepharose beads (GE) pre-washed with RIPA buffer. Affinity purification was performed at 4 °C on a nutator for 3 hours, beads were pelleted ($400 \times g$, 1 min), the supernatant removed, and the beads washed 3 times in 1 ml RIPA buffer followed by 3 times in 1 ml 50 mM ammonium bicarbonate pH 8.0 (ABC). Residual ABC was removed, and beads were resuspended in 100 μl of 50 mM ABC for protein digestion. 10 μl of a 0.1 μg/μl trypsin stock (resuspended in 20 mM Tris-HCl, pH 8) was added for a final concentration of 1 μg of trypsin and incubated at 37 °C overnight. The following day, an additional 1 μg of trypsin was added (in 10 μl of 20 mM Tris-HCl, pH 8.0) and the samples incubated an additional 2-4 hours. Beads were pelleted ($400 \times g$, 2 min) and the supernatant (peptides) transferred to a fresh 1.5 ml tube. Beads were rinsed 2 times in 100 μl HPLC water and pooled with the collected supernatant. Formic acid was added to a final concentration of 2% to end digestion (30 μl of 50% stock). The pooled supernatant was then centrifuged at $10,000 \times g$ for 10 minutes and the supernatant collected and lyophilized. Peptides were resuspended in 5% formic acid and one quarter of the sample was analyzed per MS run. 5 μl of each sample was directly loaded at 400 nl/min onto a 75 μm × 12 cm emitter packed with 3 μm ReproSil-Pur C$_{18}$-AQ (Dr. Maisch HPLC GmbH). The peptides were eluted from the column over a 90 min gradient generated by a NanoLC-Ultra 1D plus (Eksigent) nano-pump and analyzed on a TripleTOF™ 5600 instrument (AB SCIEX). The gradient was delivered at 200 nL/min starting from 2% acetonitrile with 0.1% formic acid to 35% acetonitrile with 0.1% formic acid over 90 minutes followed by a 15 min clean-up at 80% acetonitrile with 0.1% formic acid, and a 15 min re-

equilibration period in 2% acetonitrile with 0.1% formic acid for a total of 120 min. To minimize carryover between each sample, the analytical column was washed for 3 hours by running an alternating sawtooth gradient from 35% acetonitrile with 0.1% formic acid to 80% acetonitrile with 0.1% formic acid, holding each gradient concentration for 5 min. Analytical column and instrument performance were verified after each sample by loading 30 fmol BSA tryptic peptide standard (Michrom Bioresources Inc.) with 60 fmol α-Casein tryptic digest and running a short 30 min gradient. TOF MS calibration was performed on BSA reference ions before running the next sample in order to adjust for mass drift and verify peak intensity. The instrument method was set to a discovery or IDA mode which consisted of one 250 ms MS1 TOF survey scan from 400-1300 Da followed by twenty 100 ms MS2 candidate ion scans from 100-2000 Da in high sensitivity mode. Only ions with a charge of 2+ to 4+ which exceeded a threshold of 200 cps were selected for MS2, and former precursors were excluded for 10 secs after 1 occurrence. MS data generated by TripleTOF™ 5600 were stored, searched and analyzed using the ProHits laboratory information management system (LIMS) platform[90]. Within ProHits, the resulting WIFF files were first converted to an MGF format using WIFF2MGF converter and to an mzML format using ProteoWizard[91] (v3.0.4468) and the AB SCIEX MS Data Converter (V1.3 beta) and then searched using Mascot (v2.3.02) and Comet (v2012.02 rev.0). The spectra were searched with the human and adenovirus complements of the RefSeq database (version 57) from NCBI supplemented with "common contaminants" from the Max Planck Institute and the Global Proteome Machine (GPM; http://www.thegpm.org/crap/index.html). Parameters included: fully tryptic cleavages, allowing up to 2 missed cleavage sites per peptide. The mass tolerance was 40 ppm for precursors with charges of 1+ to 3+ and a tolerance of +/- 0.15 amu for fragment ions. Variable modifications were deamidated asparagine and glutamine and oxidized methionine. The results from each search engine were analyzed through TPP (the Trans-Proteomic Pipeline[92], v4.6 OCCUPY rev 3) via the iProphet pipeline[93]. Two unique peptides ions and a minimum iProphet probability of 0.95 were required for protein identification prior to analysis with SAINTexpress version 3.3[40]. Eight control runs were used for comparative purposes: 4 runs of a BioID analysis conducted on cells expressing the BirA*/FLAG tag only to control for non-specific biotinylation of intracellular proteins, and 4 runs from a BioID analysis conducted on an unrelated bait protein (EGFP) to mimic the condition in which endogenous biotinylation (which primarily occurs on mitochondrial carboxylases) would be predominant. Each negative control was analyzed in biological replicates with 4 independent biological replicates per type of control (i.e. not simple re-injections or technical replicates). A compression strategy using SAINTexpress collapsed the 8 controls to the highest 4 spectral counts for each hit, helping to capture spurious binding behavior of some contaminants. Thus, each potential prey across the 2 biological replicates of the bait is assessed for significance across the 4 highest values across the 8 controls we used. Only proteins passing a statistical threshold of FDR ≤ 0.01 were deemed high quality interactions, reported here in Supplementary Data 1.

## Purification of recombinant proteins

GST-tagged proteins were expressed in *E. coli* BL21 cells grown in LB media by induction with isopropyl-b-D-thiogalactopyranoside (IPTG) at 15 °C overnight. For NMR studies, cells were grown in minimal M9 media supplemented with ammonium-$^{15}$N chloride. Generally, cells were lysed and sonicated in 20 mM Tris-HCl (pH 7.5), 150 mM NaCl, 10% glycerol, 0.4% NP-40, protease inhibitors (Roche), 1 mM phenylmethylsulfonyl fluoride (PMSF), 10 ng/ml DNase, and 1 mM dithiothreitol (DTT). Lysate was cleared by centrifugation and incubated with glutathione resin (Amersham Pharmacia Biotech) at 4 °C for 1-2 hours. Bound proteins were eluted directly with thrombin cleavage or glutathione. Concentrated proteins were purified to homogeneity

by size exclusion chromatography using either an S75 or S200 column (GE Healthcare).

## Biochemical and biophysical protein analysis

AFDN FHA domain interactions with the SCRIB PDZ domains were measured using a MicroCal ITC200 (Malvern). Stock solutions were diluted into filtered and degassed 20 mM Tris-HCl (pH 7.5), 150 (or 20) mM NaCl and 1 mM DTT. Experiments were carried out at 25 °C. Heats of dilution were determined from control experiments in which domains were titrated into buffer alone. Data were fit using Origin 7 (Microcal). GST mixing experiments were carried out as described for immunoprecipitations, except TX100 buffer was supplemented with 10% glycerol and 1% NP-40. For interactions with small GTPases, beads were washed 5X. For in vitro mixing experiments, purified FHA or PDZ domains were added at 50 μM to glutathione beads bound to GST-fused domains.

## NMR spectroscopy

NMR data were recorded at 25 °C on a 600 MHz Bruker UltraShield spectrometer with 1.7 mm CryoProbe. NMR samples were prepared in buffer containing 20 mM Tris (pH 7.5), 100 mM NaCl, 1 mM DTT and 10% $D_2O$. Two-dimensional $^1H/^{15}N$ heteronuclear single quantum coherence (HSQC) spectra[94] were collected to analyze chemical-shift perturbations. Spectra were processed with NMRPipe[95] and analyzed using NMRView[96]. Backbone assignment of SCRIB PDZ1 was transferred from BMRBid 11207 (RIKEN), spectra that were acquired using the identical buffer conditions (20 mM Tris-HCl; 100 mM NaCl; 1 mM DTT; 0.02% NaN3).

## Phosphoproteomics

Phosphoproteomic analysis of SCRIB was performed following transient expression of N-terminally FLAG-tagged SCRIB in HEK 293T cells. Anti-FLAG antibodies were used to IP SCRIB, and precipitated proteins were run on an SDS-PAGE gel and stained with Coomassie. The SCRIB-containing band was destained in 50% MeOH (Sigma-Aldrich), shrunk in 50% acetonitrile (ACN), reconstituted in 50 mM ammonium bicarbonate with 10 mM TCEP [Tris (2-carboxyethyl) phosphine hydrochloride] (Thermo Fisher), and vortexed for 1 hr at 37 °C. Chloroacetamide (Sigma) was added for alkylation to a final concentration of 55 mM. Samples were vortexed for 1 hour at 37 °C. One microgram of trypsin was added, and digestion was performed for 8 hr at 37 °C. Peptide extraction was conducted with 90% ACN. The extracted peptide samples were dried down and solubilized in 5% ACN-0.2% formic acid (FA). The samples were loaded on a homemade $C_{18}$ precolumn (0.3-mm inside diameter [i.d.] by 5 mm) connected directly to the switching valve. They were separated on a reversed-phase column (150-μm i.d. by 150 mm) with a 56 min gradient from 10-30% ACN-0.2% FA and a 600-nl/min flow rate on a Nano-LC-Ultra-2D (Eksigent) connected to a Q-Exactive Plus (ThermoFisher). Each full MS spectrum acquired at a resolution of 70,000 was followed by 12 tandem MS-MS spectra on the most abundant multiply charged precursor ions. Tandem-MS experiments were performed using collision-induced dissociation (HCD) at a collision energy of 27%. The data were processed using PEAKS 8.5 (Bioinformatics Solutions) and a human database. Mass tolerances on precursor and fragment ions were 10 ppm and 0.01 Da, respectively. Variable selected post-translational modifications were carbamidomethyl (C), oxidation (M), deamidation (NQ), and phosphorylation (STY). The data were visualized with Scaffold (protein threshold, 99% with at least 2 peptides identified and a false-discovery rate [FDR] of 1% for peptides).

## Cell imaging

Cells were split on ethanol sterilized coverslips in 6 well dishes and incubated at 37 °C in 5% CO2 for 24 hrs. For immunostaining, cells were washed with PBS and fixed with 4% Paraformaldehyde (VWR) 48 hrs

post-transfection. Permeabilization of cells was done in PBS containing 0.05% Tween-20 (PBS-T) and blocking with 4% FBS in PBS-T. Primary antibodies were diluted in blocking reagent and incubated with coverslips for 1 hr at 37 °C in a humidified chamber. Coverslips were washed with PBS-T and incubated with secondary antibody for 1 hour at 37 °C in a humidified chamber (goat anti-mouse-Tx-Red, goat anti-rabbit-Cy5, goat anti-rabbit-Alexa Fluor 488, or Hoechst (1:2000)). Following a final wash with PBS-T and ethanol, coverslips were mounted on slides with the Prolong Diamond antifade mountant (Life Technologies) and dried for 24 hr before acquisition. Imaging of cells was performed using a laser scanning LSM-880 microscope (Zeiss). All images were taken with a 63× objective. 12 to 15 z-stacks were acquired (0.25 μm thickness) for each image and were merged by an XY orthogonal projection with the Zen lite 2.3 software (Zeiss). The following laser and detection wavelength were used: Hoechst (excitation 405 nm - detection 455/45 nm), EGFP and Alexa Fluor 488 (excitation 488 nm - detection 525/25 nm), Tx-Red (excitation 561 nm - detection 602/28 nm) and Cy5 (excitation 633 nm - detection 690/50 nm). To examine co-localization of SCRIB and AFDN we looked at z orthogonal projections in X or Y planes using ZEN Blue 2.5.

## Cell culture and immunoprecipitation

Human embryonic kidney epithelial (HEK 293T; ATCC CRL-3216), human cervix epithelial (HeLa; ATCC CCL-2), human mammary epithelial (MCF7; ATCC HTB-22) and canine kidney epithelial (MDCKII; ATCC CRL-2936) cells were maintained in Dulbecco's Modified Eagle's Medium containing 10% fetal bovine serum. For recombinant protein expression, cells were transiently transfected with PEI[97]. Stable cell lines for BioID analysis were generated as Flp-In T-REx cell pools in the human epithelial HeLa cell line. For immunoprecipitation experiments, transfected cells were lysed in TX100 buffer (20 mM Tris (pH 7.5), 150 mM NaCl, 1 mM DTT, 5 mM MgCl2 (for GTPase interactions), 1% NP-40 and 1% Triton X-100) and protease inhibitors. Lysates were cleared by centrifugation and incubated with pre-washed Protein G sepharose and immunoprecipitating antibody. Following 1-2 hour incubation, beads were washed 3X with TX100, separated by SDS-PAGE and transferred to a nitrocellulose membrane for Western blot analysis. Membranes were blocked in TBST containing 5% (wt per vol) skim milk. Primary antibodies were detected with anti-mouse Ig or anti-rabbit Ig antibodies conjugated to horseradish peroxidase followed by treatment with ECL (Pierce). Detection was done on a Bio-Rad ChemiDoc imaging system equipped with ImageLab software. To ascertain phosphatase sensitivity, 100 μL (100 U) per 1600-1800 μg of protein was added to lysates (FastAP Alkaline Phosphatase, ThermoFisher) and these were incubated at 37 °C for 1 hr. Efficiency of the treatment was verified by blotting lysates with anti-pERK.

## Generation of AFDN and SCRIB CRISPR KO cells and rescues

*AFDN* and *SCRIB* MCF7 knockout cell lines were engineered using a CRISPR/Cas9 approach with guide RNAs for *SCRIB* (5′-GGAGAGCATCAAGTTCTGCA-3′ and 5′-CTGGAGATCGCGGACTTCAG-3′, both targeting exon 3 and specifically amino acids P98-K105 and L107-S113, respectively) or *AFDN* (5′-GTTCCAGTGGTGGATGATGT-3′ targeting exon 1 and amino acids D15-N21, and 5′- AGATGTAATCGAAACGCTCG-3′ targeting exon 2 and amino acids Q69-A76). sgRNAs were designed using the online tool at https://zlab.bio/guidedesign-resources. Each guide was cloned into the pSpCas9 (BB)-2A-Puro pX459 vector (Addgene #62988). Clonal cell lines were obtained by serial dilution and isolation of colonies. Knockout was confirmed by immunoblotting. The precise nature of the CRISPR/Cas9-mediated cleavage was assessed by Sanger sequencing, following amplification of genomic DNA using primers specific for *AFDN* (fwd-ACCCATCTCTCTACATTAGTCTCAG, rev-CTGTCTCTTCACAGGTCAGGTC) or *SCRIB* (fwd-TGAGCGACAACGAGATCCAG, rev-ACCCTAGTTCCACCTGAGAAGG). This region of *AFDN* is highly GC rich and difficult to amplify, clean PCR

product could not be obtained for clone M1C3. Two additional primers (fwd-CCAGGACCATGTCGGCGGG, rev-GGCAGTACTAGAGACCCGAATA) were used in addition to the originals to amplify reversed transcribed cDNA, but this also failed to provide suitable product for sequencing. Precise edits for the four clones selected to make a reconstituted pooled population of *AFDN* KO or *SCRIB* KO are detailed in the table below, with the KO score derived from Synthego's ICE tool:

| MCF7 clone | Indel % | KO score | Indel | Genotype |
|---|---|---|---|---|
| AFDN sg2 clone-4 (**M2C4**) | 95 | 95 | −7, +1 | Added C or G causes a heterozygous indel resulting in a 39 aa or 62 aa truncated protein |
| AFDN sg1 clone-1 (**M1C1**) | 99 | 99 | −1 | Loss of A results in a 41 aa truncated protein |
| AFDN sg1 clone-3 (**M1C3**) | - | - | - | - |
| AFDN sg1 clone-6 (**M1C6**) | 91 | 85 | +1 | Added G results in a 49 aa truncated protein |
| SCRIB sg1 (**S1C6**) | 94 | 77 | −1, +1, −2 | Loss of T results in a 134 aa truncated protein; Added G in a 121 aa truncation; Loss of CT a 120aa protein |
| SCRIB sg1 (**S1C17**) | 97 | 97 | +1 | Added T results in a 121 aa truncated protein |
| SCRIB sg2 (**S2C13**) | 96 | 96 | −1 | Loss of T results in a 134 aa truncated protein |
| SCRIB sg2 (**S2C16**) | 98 | 98 | −2 | Loss of TT results in a 120 aa truncated protein |

To re-express AFDN and SCRIB in knockout cell lines, N-terminally EGFP-tagged full-length wild-type AFDN, AFDNΔFHA, full-length wild-type SCRIB or SCRIBΔPDZ1 were cloned into the pLenti-CMV-GFP-Neo plasmid (Addgene #17447) using Gibson Assembly. Lentiviruses were produced by co-transfection of vectors pMDLg/pRRE (Addgene #12251), pRSV-Rev (Addgene #12253) and pMD2-VSVG (Addgene #12259) with EGFP-AFDN or EGFP-SCRIB pLenti plasmids using calcium phosphate. HEK 293T cells were seeded in a 6-well plate 24 hrs prior to transfection in complete DMEM medium (10% FBS), then transfected with the above-mentioned transfection mixture. 24 hrs after transfection, medium was removed and replaced with fresh medium (DMEM plus 5% FBS). 48 hrs post-transfection, virus containing medium was harvested, filtered (0.45 µm) and viral titer determined by infecting *AFDN* KO and *SCRIB* KO pooled populations with a titration of AFDN and SCRIB lentivirus in the presence of polybrene (8 µg/ml). 72 hrs after infection, medium was replaced with fresh media containing 800 µg/ml G418 (Wisent, 400-130-IG) and cells were selected until 0% survival was observed in non-infected controls. The multiplicity of infection (MOI) was determined 72 hrs post-infection by comparing the percent of GFP fluorescence of infected cells to non-infected control cells. To evaluate silencing, the demethylating agent 5-Aza-2'-deoxycytidine (Sigma, A3656) was used at 20 µM and the histone deacetylase (HDAC) inhibitor Trichostatin-A (AdooQ Bioscience, A10947) at a concentration of 1 µM or 0.2 µM.

**Proliferation assay**
Cells were seeded at 1 million cells/well in a 6 well dish in complete media (10% FBS). The following day, cells were counted and diluted to 2 million cells/ml and then further diluted to $0.035 \times 10^6$ cells/ml. Diluted cells were seeded in duplicate at 3500 cells per well containing 100 µl of complete media in a 96-well microtiter plate. One microtiter plate was used for each time point. For T = 0, 100 µl of CellTiter-Glo (Promega, G7570) was added to each well and luminescence was measured using a plate reader (TECAN). For subsequent readings (T = 24 hrs, 48 hrs and 72 hrs), media was removed from the well and

replaced with 100 µl of complete media and 100 µl of CellTiter-Glo prior to measurements.

**MAPK and PI3K activity following EGF stimulation**
Cells were seeded in a 6-well dish in DMEM plus 10% FBS to achieve 50% confluency the next day. Cells were serum starved for 24 hrs in DMEM. EGF stimulation was carried out by replacing media in the 6-well dish with DMEM containing 100 ng/ml EGF (Gibco, PHG0311L) for the indicated time intervals. Cells were washed with PBS and lysed in RIPA buffer (20 mM Tris-HCl pH 7.5, 1 mM EDTA, 1% NP40, 0.5% sodium deoxycholate, 0.1% SDS, 140 mM NaCl, 2 mM DTT, 1x Sigma protease inhibitors (P8340), 1 mM PMSF, 5 nM okadaic acid and 1 mM sodium orthovanadate). Equal amounts of cleared lysate were resolved by SDS-PAGE, transferred to nitrocellulose and Western blotted for phospo-ERK1/2 or phospho-AKT, then re-probed for total ERK1/2 or total AKT.

**In vitro wound healing assay**
Confluent AFDN KO or SCRIB KO cells were serum starved overnight and 'wounded' by scratching with a P200 micropipette tip. Cells were washed 4x with PBS to remove debris and media was replaced with DMEM w/o phenol red (Wisent, 319-050-CL), containing penicillin/streptomycin/L-glutamine (Wisent, 450-202-EL) and 100 ng/ml EGF. Cells were grown in a CO2 incubator at 37 °C. Wound closure was monitored over a period of 96 hrs. Fresh medium (DMEM w/o phenol red plus pen/strep/L-glutamine and EGF) was replaced every 24 hrs. Images were captured every 24 hrs until 96 hrs using a 10X objective (Olympus) on an ORCA/ER CCD camera (Hamamatsu Photonic). Closure of the wound was quantitated using ImageJ software as the average distance moved by the leading edge over time.

**Statistics and reproducibility**
Statistical methods used to determine significance are described in the Figure Legends and were calculated using Microsoft Office and GraphPad Prism. All presented Western blots, Coomassie stained gels and microscopy images are representative of at least 3 independent experiments.

**Reporting summary**
Further information on research design is available in the Nature Research Reporting Summary linked to this article.

## Data availability
The mass spectrometry data has been deposited as a complete submission in the ProteomeXchange through partner MassIVE and assigned identifiers PXD007631 and MSV000081499 [https://massive.ucsd.edu/ProteoSAFe/dataset.jsp?task=233b894ab45849939eaae3ad460957e9], respectively. Structures 3UNN, 1WLN, 5VWK and 1X5Q are publicly available in the Protein Data Bank. Source data are provided with this paper.

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

## Acknowledgements

This work was supported by grants from the Canadian Institutes for Health Research (CIHR, to M.J.S. and A.-C.G.), the New Frontiers in Research Fund (NFRF, to M.J.S.), the National Science and Engineering Council of Canada (NSERC, to M.J.S.) and the Cancer Research Society (CRS, to M.J.S.). M.J.S. holds a Canada Research Chair in Cancer Signalling and Structural Biology and A.-C.G. in Functional Proteomics. R.K. was supported by fellowships from the Cole Foundation and the Fonds de recherche du Québec-Santé (FRQS). S.S. is supported by scholarships from the Fonds de Recherche du Québec-Nature et technologies (FRQNT) and FRQS. Phosphoproteomics analyses were performed by the Center for Advanced Proteomics Analyses, a Node of the Canadian Genomic Innovation Network that is supported by the Canadian Government through Genome Canada

## Author contributions

M.J.S. conceived and designed the project. M.G., M.J.S., V.G., S.S., R.K. and C.H.J. performed experiments and analyzed data. The manuscript was prepared by M.J.S. with input from V.G., R.K., S.S., M.G. and A-C.G.

## Competing interests

The authors declare no competing interests.

## Additional information

**Peer review information** *Nature Communications* thanks Mark Philips and other anonymous Reviewer(s) to the peer review of this work. Peer review Reports are available.

