## [Peer Review File · Nature Communications]

REVIEWER COMMENTS

Reviewer #1 (Remarks to the Author):

Manuscript title: Afadin couples RAS GTPases to the polarity rheostat Scribble

By: Marilyn Goudreault, Valérie Gagné, Chang Hwa Jo, Swati Singh, Ryan C. Killoran, Anne-Claude Gingras, and Matthew J. Smith

Comments: This is an extraordinary effort to characterize AFDN interaction network, using proximity labelling-MS approach. Authors were looking into both isoforms of AFDN as well. They found that AFDN binds to its previously unknown interactor, SCRIB, adhesion/polarity tumor suppressor protein. Both AFDN and SCRIB were found to be linked with RAS-induced cell growth and invasion. Next, they were looking into specific binding domains, involved in AFDN-SCRIBS complex and the molecular mechanism of this interaction. They found that that interaction is mediated through the first PDZ domain of SCRIB and the AFDN FHA domain. Further, the knockout experiments reveal the important role of AFDN-SCRIB interaction in MAPK and PI3K activation, and cell motility.

This data provide insight into new roles of AFDN, as not-so-well characterized RAS effector. I appreciate the quality of results, data and figures, and recommend this manuscript for publication.

Reviewer #2 (Remarks to the Author):

In the manuscript by Goudreault et al. entitled 'Afadin couples RAS GTPases to the polarity rheostat protein Scribble', the investigators employ proximity-based proteomics using afadin (AFDN) isoforms and identify the polarity protein SCRIB/Scribble as the top hit. Additional support for this interaction is obtained using in vitro binding assays and in-cell pull down experiments in which the authors find that the first PDZ domain of SCRIB and the AFDN FHA domain directly associate. AFDN has previously been shown to be an effector of the RAS and RAP GTPases and binds through its RA domains. Consistent with these earlier observations, the investigators find that KRAS co-localizes with AFDN to facilitate formation of the AFDN-SCRIB complex. Knockout of AFDN or SCRIB in MCF7 epithelial cells disrupts growth dependent MAPK and PI3K activation and inhibits cell motility. Results from these studies provide a new link between AFDN, SCRIB and RAS-mediated MAPK signaling.

1) The authors state that 'We observed in Fig. 2F that only the N-SCRIB, but not C-SCRIB construct, is able to pulldown the AFDN-FHA fragment, though the C-SCRIB contains 3 additional PDZ domains.' Have the 4 PDZ domains been compared to assess possible sequence determinants (e.g. specificity for the first PDZ domain)? Also, as the PDZ domains may interact or be involved in autoinhibitory interactions, have the investigators probed the individual PDZ domains or the 4 PDZ domains together?

2) Have the investigators compared binding of full length AFDN and SCRIB proteins with the individual domains (FHA/PDZ1) by in vitro assays? This comparison would be helpful in assessing whether the FHA and PDZ1 domains are sufficient for binding of AFDN and SCRIB, especially since the binding appears weak (~22 μ M). Although full-length AFDN is compared with AFDN delta FHA (Figure 2I), a faint band is discernible for the FHA deleted protein suggesting that other regions of the protein may be involved in the interaction. A similar comparison is also not made for the SCRIB protein (Figure 2J).

3) In Fig. 2C and 2D, IP with AFDN results in a strong SCRIB band with ectopic expression. Interestingly, this interaction looks to be retained with endogenous protein in panel 2D. However, the level of endogenous SCRIB protein pulled down looks to be much less than the GFP-tagged version despite high levels of endogenous expression. Is there a hypothesis for why this is observed?

4) The authors state 'To map the interaction site on the SCRIB PDZ1 domain we used NMR spectroscopy. BMRBID 11207 was used to assign 71% of the peaks in a $^1\text{H}/^{15}\text{N}$ -HSQC of ^{15}N -labelled SCRIB PDZ1.' It would be helpful state the conditions (pH, salt, etc) for published assignments in comparison to the NMR data collected, as the assignments are key for chemical shift mapping of the binding interaction. Please also indicate how well spectra overlay with previously determined assignments. Also, in Figure 3E, the full HSQC spectrum should be shown (with zoom as appropriate). A docked model of the interaction with the AFDN FHA domain could be generated using the NMR data but not included.

5) It is unclear whether there is a difference in the construct used in figure 3D (371-532) and Figure 3F (371-531). If they are the same then a correction is required. If they are different then the rationale for using two constructs with a difference in 1 amino acid is needed. Also, Figure 3D and 3F could be combined.

6) In Fig 3C, the minus phosphatase control for the pulldown is not shown.

7) Additional bands are observed in Fig. 3D, 3F for experiments with purified protein. Is this due to degradation or other protein contaminants? Some discussion is needed here.

8) In Fig. 4, it is stated that densitometry analysis of western blots showed an 8-fold increase in SCRIB protein pulled down with AFDN/KRAS G12V co-expression. However, this data does not appear to be included in the submission.

9) In Fig. 4D, authors show that increased SCRIB precipitation is observed with AFDN in the presence of KRAS G12V. AFDN is also shown to be immunoprecipitated with KRAS and related RAP GTPases through the RA domains. Interestingly, in panel 4D, some RAP GTPase appears to be pulled down with just FLAG-AFDN and EGFP-vector, but the same is not observed for KRAS G12V here. What is the source of the disconnect between panel 4C and 4D with regard to AFDN/KRAS G12V?

10) Similarly in Fig. 4E, a FLAG pulldown control absent of FLAG-AFDN shows modest levels of KRAS G12V immunoprecipitated. Is this due to residual SCRIB interaction with KRAS or an artifact using a FLAG-antibody?

11) Fig. 4F and 4G require reworking for presentation purposes. The Z-stacks are not very legible given the intent of showing merged co-localization of KRAS:AFDN:SCRIB. Though the figure panel is intended to show increased localization of KRAS G12V vs. KRAS WT to AFDN:SCRIB, this point is not currently made and it is unclear whether this is due to data interpretation or simply representation of the data. The investigators may want to consider alternative color mapping for better contrast of co-localization points, or a larger zoom/view for readers to see the merged fluorescence channels. Additionally, differentiation between SCRIB/AFDN merge vs. SCRIB/AFDN/KRAS merges could be useful for interpretation of the two separate complexes.

12) The same concern exists with Fig. 5E. The authors should provide a better representation for the immunofluorescence localization and internalization of AFDN. They state that expression of KRAS G12V in the SCRIB KO line induces internalization of AFDN in the figure legends, but this is not addressed in the Results proper. Rather, in Results, the authors state that AFDN is retained at sites of cell-cell contact.

13) Similar concerns also exist with Fig. 6A and 6B regarding representation of the data. In 6A, it appears that SCRIB may be less recruited to cell contacts with the FHA deletion of AFDN as compared to WT, though authors state the opposite. The trend the authors state, is more clear in the reverse scenario with AFDN and the SCRIB PDZ truncation in that there is less AFDN association at the cell contacts. Perhaps an alternative quantification representation for the co-localization of AFDN and SCRIB will better illustrate the authors' point/clarify the data.

14) Minor comment. In the Fig. 7 legend consider adding (A – C) at beginning of legend for panels A – C and (D – E) to maintain parallel structure.

15) The authors state that a signaling “defect” for the ERK MAPK/PI3K-AKT signaling cascade is induced with KO of AFDN or SCRIB. They may want to consider rewording, as the initial increase in temporal signaling indicates that signaling is not defective, but rather altered. Have the authors examined this temporal signaling effect of AFDN/SCRIB KO in the context of an activated KRAS such as KRAS G12V?

16) Statistics for pERK and pAKT is lacking to show a difference between “WT”/AFDN or SCRIB KO?

17) In Fig. 7I, consider a directional quantification of the leading-edge Golgi stain or a better zoomed-in insert to emphasize the loss of directionality in SCRIB/AFDN KO cells.

18) Mechanistic insight into how the AFSN/SCRIB effects the RAS-mediated pERK or pAKT activation would aid the discussion.

Reviewer #3 (Remarks to the Author):

The manuscript “Afadin couples RAS GTPases to the polarity rheostat Scribble” identifies a novel proximity interaction between Afadin and Scribble, using BioID coupled to mass spectrometry. A series of carefully crafted IP experiments identified that the PDZ domain of Scribble interacts with FHA domain of AFDN. Furthermore, the authors characterised the interaction between the domains using different techniques to reveal a model of how the two domains bind to each other. Since, AFDN binds to several RAS GTPases, the authors determined and characterised the interaction of activated forms of the GTPases with AFDN-SCRIB complex. Using CRISPR/Cas9 gene editing, the authors created a suite of single and double KO cell lines of AFDN and SCRIB. Using these cells lines, the authors further support a KRAS-AFDN-SCRIB complex formation and the requirement of a direct FHA-PDZ1 interaction for proper localisation of AFDN-SCRIB at cell contacts. Finally, the authors show that the loss of either AFDN or SCRIB disrupts ERK and AKT activation kinetics and cell motility in growth factor-dependent manner.

The manuscript is well written and experiments are performed with nice controls. The observations and conclusions drawn in this manuscript would help advance the field in a significant way. The data presented in the manuscript provides ample support to the conclusions drawn. I have few suggestions that would improve the manuscript:

1: BirA* is a large tag and can cause a significant level of mislocalisation when expressed in cells. Since the BioID results show some proteins from other compartments like ER/Golgi, mitochondria, etc, it would be nice if the authors can determine subcellular localisation of the AFDN-BirA* tagged fusion proteins (both isoforms).

2: Since the authors have created multiple CRISPR/Cas9 gene edited cell lines and assessed the precise nature of the edit by Sanger sequencing (line: 693), it would be recommended to show the precise edits and how they impact expression of that gene (i.e. Introduce STOP codon, impact splicing, destroy START codon, frame-shift, etc).

Reviewer #4 (Remarks to the Author):

Afadin (AFDN), a regulator of cell-cell contacts, has long been recognized as an effector of RAS and related small GTPases but little is known about the interaction or its functional consequences. AFDN is unusual among the dozen or so effectors of RAS because it possesses two tandem RA domains. Goudreault et al. set out to explore the AFDN interactome by proximity labeling and here report a comprehensive characterization of the interaction with their most robust hit, SCRIB, a tumor suppressor and polarity protein that possesses four tandem PDZ domains. They show that AFDN and SCRIB associate via a non-canonical interaction of the first PDZ domain with the forkhead associated (FHA) domain of AFDN and that the interaction is enhanced by GTP-bound KRAS that forms a ternary complex with the two polarity proteins. Conversely, they show that silencing AFDN or SCRIB changes the kinetics of growth-factor stimulated ERK and AKT signaling in MCF7 epithelial cells.

The manuscript is exceedingly well written. The authors walk the reader through not only the experiments and results but also the thinking behind them. The turboID proximity screen is well done, appropriately controlled, and clearly reported. The biochemical validation of the interaction of AFDN and SCRIB is outstanding, particularly the rigor applied to confirming the non-canonical nature of the interaction between the AFDN FHA domain with the first PDZ domain of SCRIB. The promiscuity of the AFDN RA domain(s) for RAS family proteins relative to the specificity of the RAF1 RBD is well demonstrated. These results are clear, novel, and of interest to cell biologists and are certainly worthy of reporting.

The weakest part of the study is the overinterpretation of the immunofluorescent localizations and co-localizations of KRAS, AFDN and SCRIB.

Fig. 2A. Here AFDN and SCRIB, both epitope-tagged, are overexpressed in a HeLa cell that is processed for immunofluorescence. It is not stated if the GFP-AFDN is imaged with the intrinsic fluorescence of GFP or if an anti-GFP antibody is employed along with the anti-FLAG antibody. No control proteins are employed nor is a control with first antibody omitted shown. The localization is indeterminate and uninformative. A single cell is shown such that one cannot determine if this represents the predominant fluorescent pattern. Since tagged, ectopically expressed proteins are used, including one tagged with GFP, it is not clear why the authors did not use mCherry-SCRIB such that they could colocalize the two proteins in live cells, which allow for more precise subcellular localization free of fixation and permeabilization artifacts.

Fig. 4F,G. In describing these micrographs the authors state on p. 9 that “tagged, wild-type KRAS does not significantly alter AFDN or SCRIB localization and does not co-localize with these proteins (Figure 4F and S4B). ... In contrast, expression of KRAS-G12V markedly disrupted cell-cell contacts and was noticeably co-localized with endogenous AFDN and SCRIB, as determined by z-plane projections (Figure 4G).” In the discussion on p. 13 the authors write that “we show that RAS-G12V is co-localized with AFDN and SCRIB at sites of cell contact, while wild-type RAS is distributed more generally across the plasma membrane.” The data do not support these conclusions. First, not shown is any disruption of cell-cell contacts in cells expressing KRAS-G12V. Three GFP-KRAS4B-G12V expressing cells are shown with three levels of expression and perhaps different z-planes. Two of these three maintain robust cell-cell contacts as determined by morphology and SCRIB and AFDN staining of areas of cell contact (anti-ZO-1 staining would be a way to look at this without imaging the experimental proteins themselves). Not shown are any of the detached cells to which the authors refer as having lost cell-cell contacts as a function of oncogenic KRAS. Second, and more important, the data show that both WT-KRAS and KRAS-G12V decorate the basolateral membrane. Indeed, both the WT and mutant KRAS decorate the entire plasma membrane (PM) as is well established in a vast literature. As expected for cell adhesion proteins imaged in confluent epithelial cells, SCRIB and AFDN decorate primarily the basolateral membrane. The conclusion that these proteins are colocalized to a greater extent with KRAS-G12V than with WT KRAS is not supported by the data shown and contrary to a vast literature on KRAS localization. Some of the problem is semantic; the concept of co-localization is somewhat ambiguous. There is co-localization on the PM in some regions of the cell but not others. This should not necessarily be interpreted as one protein pulling another to a region of PM since the same pattern would be observed if the localizations are true but unrelated to the direct interactions of the proteins. The three proteins colocalize at the basolateral membrane but not the apical membrane and this is not affected by the GTP-binding state of KRAS. Were this localization of mutant KRAS to differ from that of WT it would be contrary to a vast literature on KRAS localization that in total demonstrates that the subcellular localization driven by the prenylated HVR is not affected by the GTP-binding state. Current paradigms of RAS signaling hold the PM localization of KRAS is constitutive and it is effectors that are drawn to RAS (e.g. translocation of RAF) not vice versa. Are the authors arguing that in this case the converse is true and that the localization of KRAS is driven by that of its effector?

Fig. 5D. The altered localization of endogenous SCRIB as a consequence of silencing AFDN is described on p. 10 as “dispersed throughout the cytoplasm.” But unlike the distribution of GFP that is homogeneous and clearly cytosolic, that of SCRIB is punctate consistent with a vesicular localization and should be described as such. Co-localization with Texas-red transferrin would allow an assessment as to whether these are endosomes. Caution must be taken in what is used for permeabilization of the cells (here 0.05% Tween-20) as this can alter the appearance of vesicles. It would be wise to also try 0.1% saponin.

Despite binding of several RAS family small GTPases, in their colocalization studies the authors restricted their analysis to KRAS4B. This is unfortunate. It would be informative to also study a RAS-related binding partner that is not normally localized exclusively to the PM. RAP2 fits the bill as the authors show strong binding to AFDN and this small GTPase has been localized to endomembrane (PMID: 1923507 and 19061864). It would also be interesting to determine if RHEB interacts with AFDN since this RAS family small GTPase is expressed on lysosomes.

Minor points:

Fig. 2B. This figure sets up indirect immunofluorescence (iIF) staining of endogenous AFDN and SCRIB, which is used extensively throughout the paper. The specificity of antibodies used for iIF must always be validated by knockdown of the protein of interest, which is accomplished in Fig. 5 and this should be added to the legend of Fig. 2B.

Fig. 4D. The authors write on p. 7 that with an n=5 they saw an 8-fold enhancement of SCRIB co-IP with AFDN upon expression of KRAS4B-G12V, but they do not report results for RAP1B or RAP2C, which appear to also induce some enhancement, albeit to a lesser extent (KRAS4B>>RAP2C> RAP1B). It would be informative to report the results for each of the interacting GTPases.

Fig. S6B. It is very difficult to see the AFDN staining.

Fig. 6C,D. These z projections are convincing that true colocalization of AFDN and SCRIB requires PDZ1 and FHA, but would be even more so if they were subjected to analysis with Pearson’s coefficient.

The authors refer to KRAS throughout but they mean KRAS4B. They do not study KRAS4A. Since these splice variants differ only in their HVRs that direct subcellular trafficking this should be acknowledged.

To be a true effector of a small GTPase, three conditions must be met. The effector must bind directly to the GTPase, the binding must depend on GTP-loading of the GTPase, and the binding must in some way change the conformation or activity of the effector. RAF and HK1 meet all of these criteria but the third has been lacking for AFDN. In Fig. 4D the authors establish for the first time a change in the properties of AFDN induced by KRAS, confirming that AFDN is a bone fide effector. This should be discussed.

Because AFDN is unique in possessing tandem RA domains that, in principal, could bind two GTPases the authors have a unique opportunity to ask if either or both are required for the effect of KRAS seen in Fig. 4D. Indeed, it would be interesting to determine if RAS binds to one and RAP2 to the other RA domain.

The change in kinetics of ERK and AKT activation downstream of EGF signaling upon silencing AFDN or SCRIB shown in Fig. 7D,E is interesting but the authors do not comment on possible mechanisms. Interestingly they parallel the differential effects of NGF versus EGF in PC12 cells where only the former induces sustained ERK activation (PMID 7834738).

This revised manuscript addresses all Referee comments with either experimental data or further clarification, as required. We have added the following results to the manuscript:

- Purified SCRIB PDZ domains 2 and 3-4, measured binding to the AFDN FHA domain (no binding demonstrates specificity of the AFDN domain for SCRIB PDZ1)
- Determined GTPase specificity of AFDN RA domains 1 and 2 alone, to contrast what was observed with the RA1-2 construct (GTPase binding is avidity driven by the tandem domains, particularly KRAS)
- Derived suitable conditions for our phosphatase treated co-IP of AFDN and SCRIB, and included the untreated control
- Demonstrated that AFDN and RAS GTPases co-localize in HeLa cells, which do not express detectable levels of endogenous AFDN, substantiating AFDN as a RAS effector
- Obtained new images for KRAS expressed in MCF7 cells to demonstrate how cells expressing activated KRAS detach from the monolayer
- Used RAB5, RAB7 and RAB11 GTPases to resolve whether the punctate SCRIB pattern observed in AFDN KO MCF7 cells is localized to endosomes
- Demonstrated co-localization of AFDN with activated RAP1B and RAP2C GTPases

Specific comments to each reviewer follow.

Reviewer #1

Comments: This is an extraordinary effort to characterize AFDN interaction network, using proximity labelling-MS approach. Authors were looking into both isoforms of AFDN as well. They found that AFDN binds to its previously unknown interactor, SCRIB, adhesion/polarity tumor suppressor protein. Both AFDN and SCRIB were found to be linked with RAS-induced cell growth and invasion. Next, they were looking into specific binding domains, involved in AFDN-SCRIBS complex and the molecular mechanism of this interaction. They found that that interaction is mediated through the first PDZ domain of SCRIB and the AFDN FHA domain. Further, the knockout experiments reveal the important role of AFDN-SCRIB interaction in MAPK and PI3K activation, and cell motility.

This data provide insight into new roles of AFDN, as not-so-well characterized RAS effector. I appreciate the quality of results, data and figures, and recommend this manuscript for publication.

We thank the reviewer for their positive comments and appreciation for our work.

Reviewer #2

In the manuscript by Goudreault et al. entitled 'Afadin couples RAS GTPases to the polarity rheostat protein Scribble', the investigators employ proximity-based proteomics using afadin (AFDN) isoforms and identify the polarity protein SCRIB/Scribble as the top hit. Additional support for this interaction is obtained using in vitro binding assays and in-cell pull down experiments in which the authors find that the first PDZ domain of SCRIB and the AFDN FHA domain directly associate. AFDN has previously been shown to be an effector of the RAS and RAP GTPases and binds through its RA domains. Consistent with these earlier observations, the investigators find that KRAS co-localizes with AFDN to facilitate formation of the AFDN-SCRIB

complex. Knockout of AFDN or SCRIB in MCF7 epithelial cells disrupts growth dependent MAPK and PI3K activation and inhibits cell motility. Results from these studies provide a new link between AFDN, SCRIB and RAS-mediated MAPK signaling.

We thank the reviewer for their thoughtful suggestions and have attempted to address all points raised with comments, or by acquiring new experimental data.

1) The authors state that ‘We observed in Fig. 2F that only the N-SCRIB, but not C-SCRIB construct, is able to pulldown the AFDN-FHA fragment, though the C-SCRIB contains 3 additional PDZ domains.’ Have the 4 PDZ domains been compared to assess possible sequence determinants (e.g. specificity for the first PDZ domain)? Also, as the PDZ domains may interact or be involved in autoinhibitory interactions, have the investigators probed the individual PDZ domains or the 4 PDZ domains together?

There are significant literature describing efforts to determine PDZ domain binding specificity. To date, it has not been possible to establish binding specificity of individual domains based on sequence alone. The most valuable approaches have been large scale, non-biased screens of PDZ binding to peptide libraries and many include the PDZ domains of SCRIB¹⁻³. Indeed, several recent papers have resolved SCRIB PDZ 1-4 specificity for individual substrates, though resolving specificity determinants within the domains themselves has required structural elucidation⁴⁻⁶. As we have been unable to successfully crystallize the PDZ1-FHA complex, and most library screens to date have focused on C-terminal PDZ motifs, a sequence-based prediction of why PDZ1 binds to AFDN while PDZ2, 3, and 4 domains of SCRIB do not is currently not feasible. To satisfy the reviewers request and demonstrate that our elucidated interaction is specific to PDZ1, we generated expression vectors for SCRIB PDZ2 and the PDZ3-4 supramodule⁷. We expressed and purified these domains to homogeneity and used ITC to measure their binding affinity for the AFDN FHA domain (new **Supplementary Fig. 3d**). We observed no binding between the FHA domain and SCRIB PDZ2, while the PDZ3-4 module showed only extremely weak binding on the order of ~100 mM. Thus, the AFDN-SCRIB interaction is highly specific for the first PDZ of SCRIB, though the determinants of this specificity must be elucidated with future structural characterization of this complex.

1. Zhang, Y. et al. Convergent and divergent ligand specificity among PDZ domains of the LAP and zonula occludens (ZO) families. *Journal of Biological Chemistry* 281, 22299–22311 (2006).
2. Ivarsson, Y. et al. Large-scale interaction profiling of PDZ domains through proteomic peptide-phage display using human and viral phage peptidomes. *Proceedings of the National Academy of Sciences of the United States of America* **111**, 2542–7 (2014).
3. Mu, Y., Cai, P., Hu, S., Ma, S. & Gao, Y. Characterization of diverse internal binding specificities of PDZ domains by yeast two-hybrid screening of a special peptide library. *PLoS ONE* **9**, e88286 (2014).
4. Lim, K. Y. B., Gödde, N. J., Humbert, P. O. & Kvangsakul, M. Structural basis for the differential interaction of Scribble PDZ domains with the guanine nucleotide exchange factor β -PIX. *Journal of Biological Chemistry* **292**, 20425–20436 (2017).

5. Caria, C. et al. Structural analysis of phosphorylation-associated interactions of MCC to Scribble PDZ domains. *FEBS Journal* **286**, 4910-4925 (2019).
6. How, J. Y., Stephens, R. K., Lim, K. Y. B., Humbert, P. O. & Kvensakul, M. Structural basis of the human Scribble–Vangl2 association in health and disease. *Biochemical Journal* **478**, 1321–1332 (2021).
7. Ren, J. et al. Interdomain interface-mediated target recognition by the Scribble PDZ34 supramodule. *Biochemical Journal* **468**, 133–144 (2015).

2) *Have the investigators compared binding of full length AFDN and SCRIB proteins with the individual domains (FHA/PDZ1) by in vitro assays? This comparison would be helpful in assessing whether the FHA and PDZ1 domains are sufficient for binding of AFDN and SCRIB, especially since the binding appears weak (~22 μ M). Although full-length AFDN is compared with AFDN delta FHA (Figure 2I), a faint band is discernible for the FHA deleted protein suggesting that other regions of the protein may be involved in the interaction. A similar comparison is also not made for the SCRIB protein (Figure 2J).*

While we would be happy to measure *in vitro* binding of full length SCRIB and AFDN, this is an extremely challenging experiment. Both proteins are ~200 kDa in size and cannot be expressed and purified from bacteria. We have attempted to purify these proteins using a mammalian cell expression system, but AFDN in particular has a propensity to aggregate *in vitro* as a full length protein. While we anticipate purifying and studying this complex in the future, it is beyond the scope of this manuscript and will require many years of work. I do believe the reviewer is correct that there are likely additional contact points outside PDZ1-FHA. The two proteins co-IP under highly stringent conditions (even in RIPA buffer), indicating the measured *in vitro* affinity between the PDZ1 and FHA domains (5-15 μ M depending on NaCl concentration) is perhaps insufficient to rationalize the overall robust binding of the two proteins from cell lysate. It is possible that the AFDN oligomers we observe following its purification from mammalian cells are genuine, perhaps supporting a model whereby multiple AFDN proteins can strongly co-precipitate SCRIB through multiple FHA-PDZ1 interactions of moderate affinity.

3) *In Fig. 2C and 2D, IP with AFDN results in a strong SCRIB band with ectopic expression. Interestingly, this interaction looks to be retained with endogenous protein in panel 2D. However, the level of endogenous SCRIB protein pulled down looks to be much less than the GFP-tagged version despite high levels of endogenous expression. Is there a hypothesis for why this is observed?*

There are numerous explanations for why co-immunoprecipitation of endogenous proteins may not re-capitulate the strong complex observed by ectopic expression. A primary factor is the binding of antibodies to the proteins directly, rather than to a recombinant tag. In this case, the AFDN polyclonal antibodies were raised against a “synthetic peptide corresponding to Mouse Afadin aa 1800 to the C-terminus (abcam)”. While this should not disrupt the PDZ1-FHA interaction, it is difficult to predict if antibody binding to the C-terminus of AFDN may disrupt SCRIB binding until we have a structure of the full proteins in complex. Indeed, related to the previous question, we now believe that AFDN oligomerization is mediated through its C-terminal

region and it is possible that disruption of this higher order structure diminishes the amount of SCRIB bound in an IP. Moreover, it is conceivable that a fraction of endogenous SCRIB/AFDN in these cells are post-translationally modified in a manner that may disrupt their interaction, or that a population of the endogenous proteins are engaged in alternative complexes that are competitive with the AFDN-SCRIB module. Altogether, we were happy to observe the complex could be validated with endogenous proteins and future work will resolve the detailed mechanisms by which it is regulated in cells.

4) The authors state ‘To map the interaction site on the SCRIB PDZ1 domain we used NMR spectroscopy. BMRBID 11207 was used to assign 71% of the peaks in a 1H/15N-HSQC of 15N-labelled SCRIB PDZ1.’ It would be helpful state the conditions (pH, salt, etc) for published assignments in comparison to the NMR data collected, as the assignments are key for chemical shift mapping of the binding interaction. Please also indicate how well spectra overlay with previously determined assignments. Also, in Figure 3E, the full HSQC spectrum should be shown (with zoom as appropriate). A docked model of the interaction with the AFDN FHA domain could be generated using the NMR data but not included.

To address the first question, the assignment of SCRIB PDZ1 in the BMRB was deposited by RIKEN and there is no accompanying manuscript, but the buffer conditions in the PDB are:

20mM d-Tris-HCl; 100mM NaCl; 1mM d-DTT; 0.02% NaN₃

This is nearly identical to the conditions used to acquire our own PDZ1 spectra:

20 mM Tris (pH 7.5), 100 mM NaCl, 1 mM DTT

To further indicate how well the previously assigned spectra overlays with our own data, we have added the full spectra superimposed with projected peaks from the BMRB assignment (new **Supplementary Fig. 4a**). This clearly demonstrates how similar our experimental data is to the previously derived assignment, and how we were able to unambiguously assign 70% of the observed peaks. The second point raised here is to include a full HSQC of the AFDN FHA domain in **Fig. 3e**. We do not believe a full HSQC would offer any additional information, and the space constraints make it difficult to fit complete spectra in **Fig. 3**. The panels shown demonstrate how the PDZ1 domain induces chemical shift perturbations in residues located in the C-terminal extension of the FHA domain. There are no chemical shifts from these residues in the broader spectrum, so we have chosen to leave this figure as originally presented. Finally, with regards to modeling we have used HADDOCK to generate models of the PDZ1-FHA interaction, aided by chemical shift perturbations used as restraints. While the models are interesting, they are not experimentally corroborated and do not provide any additional value to the current work. We prefer to wait for a proper structural elucidation of the full complex rather than speculate on a predicted model.

5) It is unclear whether there is a difference in the construct used in figure 3D (371-532) and Figure 3F (371-531). If they are the same then a correction is required. If they are different then the rationale for using two constructs with a difference in 1 amino acid is needed. Also, Figure 3D and 3F could be combined.

These are the same construct (371-532) and the labelling has been corrected.

6) *In Fig 3C, the minus phosphatase control for the pulldown is not shown.*

We have redone this experiment and now include a minus phosphatase control in **Fig. 3c**.

7) *Additional bands are observed in Fig. 3D, 3F for experiments with purified protein. Is this due to degradation or other protein contaminants? Some discussion is needed here.*

The proteins used in these binding assays are GST-tagged and bound to glutathione beads. The smaller bands beneath our proteins of interest are typical degradation bands observed with GST fusion proteins, particularly following incubation with the partner protein (here, purified PDZ1) on a nutator for >1 hour followed by multiple washes.

8) *In Fig. 4, it is stated that densitometry analysis of western blots showed an 8-fold increase in SCRIB protein pulled down with AFDN/KRAS G12V co-expression. However, this data does not appear to be included in the submission.*

We have added quantitation of SCRIB co-immunoprecipitation with AFDN in the new **Fig. 4g**. The plot shows SCRIB levels when co-expressed with KRAS-G12V, as well as RAP1B-G12V and RAP2C-G12V. Ratios are in comparison to SCRIB levels co-precipitated when AFDN is expressed with wild-type KRAS.

9) *In Fig. 4D, authors show that increased SCRIB precipitation is observed with AFDN in the presence of KRAS G12V. AFDN is also shown to be immunoprecipitated with KRAS and related RAP GTPases through the RA domains. Interestingly, in panel 4D, some RAP GTPase appears to be pulled down with just FLAG-AFDN and EGFP-vector, but the same is not observed for KRAS G12V here. What is the source of the disconnect between panel 4C and 4D with regard to AFDN/KRAS G12V?*

(Previous Fig. 4d is now Fig. 4e) We show multiple times that KRAS-G12V can be precipitated from cell lysate by the AFDN RA domains (**Figs 2f and 4c/d**) or that KRAS co-precipitates with full length AFDN (**Figs 4a/b, 5c and Supplementary Fig. 5a**). We have also previously solved a crystal structure of the AFDN RA1 bound to RAS-GTP and characterized this complex by ITC (*Nat. Comm.*, 2017). These data clearly demonstrate that KRAS-G12V binds AFDN, though a strong band is not visible in lane 9 of **Fig. 4e**. This is likely because this is a triple transient transfection (3 plasmids) and the levels of GTPase expressed are relatively low. This might be improved by generating series of stable cell lines expressing the GTPases, but the objective of this experiment was to determine the relative levels of SCRIB present in AFDN co-IPs when the activated GTPases are present. As it worked well for this, it did not seem necessary to further resolve the KRAS-AFDN interaction when it is studied extensively in other experiments.

10) Similarly in Fig. 4E, a FLAG pulldown control absent of FLAG-AFDN shows modest levels of KRAS G12V immunoprecipitated. Is this due to residual SCRIB interaction with KRAS or an artifact using a FLAG-antibody?

(Previous **Fig. 4e** is now **Fig. 4f**) These are merely background bands common when working with RAS GTPases. The proteins are unstable when nucleotide free or when prenylated in the absence of lipid membrane and it is common to have background. As in the comment above, this is a triple transient transfection and the expression level of the GTPases is low (we must expose for ~2 minutes to observe strong bands). The objective of this experiment was to resolve whether SCRIB Δ PDZ1 alters the interaction between AFDN and activated KRAS, and the experiment clearly demonstrates that it does not.

11) Fig. 4F and 4G require reworking for presentation purposes. The Z-stacks are not very legible given the intent of showing merged co-localization of KRAS:AFDN:SCRIB. Though the figure panel is intended to show increased localization of KRAS G12V vs. KRAS WT to AFDN:SCRIB, this point is not currently made and it is unclear whether this is due to data interpretation or simply representation of the data. The investigators may want to consider alternative color mapping for better contrast of co-localization points, or a larger zoom/view for readers to see the merged fluorescence channels. Additionally, differentiation between SCRIB/AFDN merge vs. SCRIB/AFDN/KRAS merges could be useful for interpretation of the two separate complexes.

We have taken multiple approaches to address the points raised here. First, we enlarged the z-stack panels in these images to make the expression patterns more legible (now **Fig. 5a/b**). More importantly, we reconsidered whether MCF7 cells were an appropriate system to demonstrate co-localization of RAS GTPases with AFDN, as both proteins are constitutively membrane localized in polarized epithelial cells. This made it very challenging to demonstrate co-localization driven by RAS activation. Instead, we went back to the HeLa cell system as this line does not express detectable levels of endogenous AFDN and does not form adherens junctions. The new **Fig. 5c** shows how expression of activated KRAS, and not wild-type KRAS, recruits exogenously expressed AFDN to the cell membrane. We hope the reviewer agrees this is a more robust demonstration that AFDN co-localizes with activated RAS in cells and that it should be considered a RAS effector.

12) The same concern exists with Fig. 5E. The authors should provide a better representation for the immunofluorescence localization and internalization of AFDN. They state that expression of KRAS G12V in the SCRIB KO line induces internalization of AFDN in the figure legends, but this is not addressed in the Results proper. Rather, in Results, the authors state that AFDN is retained at sites of cell-cell contact.

In response to this point, and a related point from Reviewer 4, we have added numerous images of immunostained AFDN in KRAS-expressing MCF7 cells (the new **Supplementary Fig. 5c/d**). These images show that AFDN is no longer visible at cell-cell contacts in the majority of cells expressing KRAS-G12V, while it is retained at cell contacts in those expressing wild-type KRAS. Indeed, most cells expressing KRAS-G12V are no longer associated with adjacent cells and empty

space is clearly observable surrounding these cells. The results are appropriately discussed in the Results section.

13) Similar concerns also exist with Fig. 6A and 6B regarding representation of the data. In 6A, it appears that SCRIB may be less recruited to cell contacts with the FHA deletion of AFDN as compared to WT, though authors state the opposite. The trend the authors state, is more clear in the reverse scenario with AFDN and the SCRIB PDZ truncation in that there is less AFDN association at the cell contacts. Perhaps an alternative quantification representation for the co-localization of AFDN and SCRIB will better illustrate the authors' point/clarify the data.

We have calculated Pearson's coefficients to quantify the levels of endogenous SCRIB co-localized with EGFP-tagged wild-type AFDN or the Δ FHA variant (in AFDN KO cells), or endogenous AFDN with EGFP-tagged wild-type SCRIB or its Δ PDZ1 variant (in SCRIB KO cells). These calculations were performed on $n \geq 8$ images similar to those presented in **Fig. 7c/d**. The results are presented in the new **Supplementary Fig. 7c**. This quantitation resolved a significant difference in the co-localization of endogenous SCRIB/AFDN with the respective wild-type rescues compared with domain deletions, and is now presented in the text.

14) Minor comment. In the Fig. 7 legend consider adding (A – C) at beginning of legend for panels A – C and (D – E) to maintain parallel structure.

This was corrected in the figure legend.

15) The authors state that a signaling “defect” for the ERK MAPK/PI3K-AKT signaling cascade is induced with KO of AFDN or SCRIB. They may want to consider rewording, as the initial increase in temporal signaling indicates that signaling is not defective, but rather altered. Have the authors examined this temporal signaling effect of AFDN/SCRIB KO in the context of an activated KRAS such as KRAS G12V?

This may be semantics, but ‘defect’ is defined as “an imperfection or abnormality that impairs quality, function, or utility”. We would argue that the disrupted kinetics of AKT and ERK activation demonstrated here represent a genuine abnormality in response to EGF stimulation and can thus be appropriately described as a “defect”. See also the response to Reviewer 4, point 12.

16) Statistics for pERK and pAKT is lacking to show a difference between “WT”/AFDN or SCRIB KO?

I believe the reviewer is referring to **Fig. 8d/e** (previously **Fig. 7d/e**), which graphs quantitation of pERK and pAKT in the parental cell line vs the AFDN and SCRIB KO lines. This is a time course of EGF stimulation, and to my knowledge there is no established statistical approach that will determine a p-value across the entire series. A 2-way ANOVA can be used, but this would treat the time points randomly rather than sequential. If the course resulted in a linear response, we could solve this with regression, but the response is not linear. We can use multiple independent t-tests (perhaps focused on the time points of interest), but this disregards that the series are

sequential and therefore does not provide value. We have left the qualitative graphs in **Fig. 8d/e** without p-values for now, but if the reviewer has insights to how such non-linear time course data can be tested statistically, we are happy to perform the analysis.

17) In Fig. 7I, consider a directional quantification of the leading-edge Golgi stain or a better zoomed-in insert to emphasize the loss of directionality in SCRIB/AFDN KO cells.

We have added a zoomed image of cells at the leading edge to **Fig. 8i** (previously **Fig. 7i**).

18) Mechanistic insight into how the AFSN/SCRIB effects the RAS-mediated pERK or pAKT activation would aid the discussion.

Please see our answer to Reviewer 4, point 12.

Reviewer #3

The manuscript “Afadin couples RAS GTPases to the polarity rheostat Scribble” identifies a novel proximity interaction between Afadin and Scribble, using BioID coupled to mass spectrometry. A series of carefully crafted IP experiments identified that the PDZ domain of Scribble interacts with FHA domain of AFDN. Furthermore, the authors characterised the interaction between the domains using different techniques to reveal a model of how the two domains bind to each other. Since, AFDN binds to several RAS GTPases, the authors determined and characterised the interaction of activated forms of the GTPases with AFDN-SCRIB complex. Using CRISPR/Cas9 gene editing, the authors created a suite of single and double KO cell lines of AFDN and SCRIB. Using these cells lines, the authors further support a KRAS-AFDN-SCRIB complex formation and the requirement of a direct FHA-PDZ1 interaction for proper localisation of AFDN-SCRIB at cell contacts. Finally, the authors show that the loss of either AFDN or SCRIB disrupts ERK and AKT activation kinetics and cell motility in growth factor-dependent manner. The manuscript is well written and experiments are performed with nice controls. The observations and conclusions drawn in this manuscript would help advance the field in a significant way. The data presented in the manuscript provides ample support to the conclusions drawn. I have few suggestions that would improve the manuscript:

1) BirA is a large tag and can cause a significant level of mislocalisation when expressed in cells. Since the BioID results show some proteins from other compartments like ER/Golgi, mitochondria, etc, it would be nice if the authors can determine subcellular localisation of the AFDN-BirA* tagged fusion proteins (both isoforms).*

We thank the reviewer for their helpful comments and will address the points raised here. A distinct advantage of proximity-based proteomics is the insight provided to subcellular localization. The results presented in **Fig. 1** (and accompanying **Table 1**) make clear that BirA*-AFDN is properly behaved, as the majority of identified preys are plasma membrane proteins involved in cell adhesion (including most previously known AFDN interactors). The reviewer is correct that the

BirA* tag is large, and to satisfy the request we imaged our BirA*-FLAG-AFDN iso1 construct in the MCF7 cell line:

This confocal image shows that BirA*-FLAG-AFDN is prominently localized at cell-cell contacts, as expected. We include here staining with Streptavidin-488 following 2 hours of incubation with biotin. This shows the majority of biotinylated proteins are also at the plasma membrane, though AFDN and Streptavidin-488 signals are also visible in the interior of these cells (Strep-488 also characteristically marks mitochondria in the surrounding cells). We are confident based on these images and the derived proteomic data that BirA*-AFDN appropriately localizes in cells.

2) Since the authors have created multiple CRISPR/Cas9 gene edited cell lines and assessed the precise nature of the edit by Sanger sequencing (line: 693), it would be recommended to show the precise edits and how they impact expression of that gene (i.e. Introduce STOP codon, impact splicing, destroy START codon, frame-shift, etc).

We have added a table to the Results section (under the subheading *Generation of AFDN and SCRIB CRISPR KO cells and Rescues*) that describes sequencing of the KO cell lines in detail.

Reviewer #4

Afadin (AFDN), a regulator of cell-cell contacts, has long been recognized as an effector of RAS and related small GTPases but little is known about the interaction or its functional consequences. AFDN is unusual among the dozen or so effectors of RAS because it possesses two tandem RA domains. Goudreault et al. set out to explore the AFDN interactome by proximity labeling and here report a comprehensive characterization of the interaction with their most robust hit, SCRIB, a tumor suppressor and polarity protein that possesses four tandem PDZ domains. They show that AFDN and SCRIB associate via a non-canonical interaction of the first PDZ domain with the forkhead associated (FHA) domain of AFDN and that the interaction is enhanced by GTP-bound KRAS that forms a ternary complex with the two polarity proteins. Conversely, they show that silencing AFDN or SCRIB changes the kinetics of growth-factor stimulated ERK and AKT signaling in MCF7 epithelial cells. The manuscript is exceedingly well written. The authors walk the reader through not only the experiments and results but also the thinking behind them. The

turboID proximity screen is well done, appropriately controlled, and clearly reported. The biochemical validation of the interaction of AFDN and SCRIB is outstanding, particularly the rigor applied to confirming the non-canonical nature of the interaction between the AFDN FHA domain with the first PDZ domain of SCRIB. The promiscuity of the AFDN RA domain(s) for RAS family proteins relative to the specificity of the RAF1 RBD is well demonstrated. These results are clear, novel, and of interest to cell biologists and are certainly worthy of reporting. The weakest part of the study is the overinterpretation of the immunofluorescent localizations and co-localizations of KRAS, AFDN and SCRIB.

We thank the reviewer for evaluating our manuscript and will address the concerns here, including with the addition of new experimental data.

1) Fig. 2A. Here AFDN and SCRIB, both epitope-tagged, are overexpressed in a HeLa cell that is processed for immunofluorescence. It is not stated if the GFP-AFDN is imaged with the intrinsic fluorescence of GFP or if an anti-GFP antibody is employed along with the anti-FLAG antibody. No control proteins are employed nor is a control with first antibody omitted shown. The localization is indeterminant and uninformative. A single cell is shown such that one cannot determine if this represents the predominant fluorescent pattern. Since tagged, ectopically expressed proteins are used, including one tagged with GFP, it is not clear why the authors did not use mCherry-SCRIB such that they could colocalize the two proteins in live cells, which allow for more precise subcellular localization free of fixation and permeabilization artifacts.

In **Fig. 2a** we show the localization of ectopically tagged AFDN and SCRIB in HeLa cells (as this was the cell line from which our proteomics data was collected) before moving on to MCF7 cells. The reviewer states the localization is ‘*indeterminant and uninformative*’, but this is indeed the diffuse localization pattern we observe from these proteins in this epithelial line, which lack defined cell-cell contacts and apical-basal polarity. Levels of endogenous AFDN in HeLa cells were too low to detect by immunofluorescence. Endogenous SCRIB is expressed, can be detected, and displays a diffuse/punctate localization when immunostained that is highly reminiscent of SCRIB in *AFDN* KO MCF7 cells:

Others have observed this pattern in different cell types¹. To satisfy the reviewers request, we co-expressed EGFP-AFDN and Cherry-SCRIB in these cells (the new **Supplementary Fig. 1d**), or appropriate controls, to avoid fixation/permeabilization effects. We again observe broad staining in the cytoplasm for both AFDN and SCRIB, with some weak concentration at the cell cortex. In

the following section we describe how this was used to improve our understanding of RAS-mediated recruitment of AFDN to the plasma membrane, something that was more difficult in the polarized MCF7 cell line where AFDN and SCRIB are characteristically membrane-proximal even in the absence of activated GTPases.

1. Anastas, J. N. et al. A protein complex of SCRIB, NOS1AP and VANGL1 regulates cell polarity and migration, and is associated with breast cancer progression. *Oncogene* 31, 3696–3708 (2012).

2) *Fig. 4F,G. In describing these micrographs the authors state on p. 9 that “tagged, wild-type KRAS does not significantly alter AFDN or SCRIB localization and does not co-localize with these proteins (Figure 4F and S4B). ... In contrast, expression of KRAS-G12V markedly disrupted cell-cell contacts and was noticeably co-localized with endogenous AFDN and SCRIB, as determined by z-plane projections (Figure 4G).” In the discussion on p. 13 the authors write that “we show that RAS-G12V is co-localized with AFDN and SCRIB at sites of cell contact, while wild-type RAS is distributed more generally across the plasma membrane.” The data do not support these conclusions. First, not shown is any disruption of cell-cell contacts in cells expressing KRAS-G12V. Three GFP-KRAS4B-G12V expressing cells are shown with three levels of expression and perhaps different z-planes. Two of these three maintain robust cell-cell contacts as determined by morphology and SCRIB and AFDN staining of areas of cell contact (anti-ZO-1 staining would be a way to look at this without imaging the experimental proteins themselves). Not shown are any of the detached cells to which the authors refer as having lost cell-cell contacts as a function of oncogenic KRAS. Second, and more important, the data show that both WT-KRAS and KRAS-G12V decorate the basolateral membrane. Indeed, both the WT and mutant KRAS decorate the entire plasma membrane (PM) as is well established in a vast literature. As expected for cell adhesion proteins imaged in confluent epithelial cells, SCRIB and AFDN decorate primarily the basolateral membrane. The conclusion that these proteins are colocalized to a greater extent with KRAS-G12V than with WT KRAS is not supported by the data shown and contrary to a vast literature on KRAS localization. Some of the problem is semantic; the concept of co-localization is somewhat ambiguous. There is co-localization on the PM in some regions of the cell but not others. This should not necessarily be interpreted as one protein pulling another to a region of PM since the same pattern would be observed if the localizations are true but unrelated to the direct interactions of the proteins. The three proteins colocalize at the basolateral membrane but not the apical membrane and this is not affected by the GTP-binding state of KRAS. Were this localization of mutant KRAS to differ from that of WT it would be contrary to a vast literature on KRAS localization that in total demonstrates that the subcellular localization driven by the prenylated HVR is not affected by the GTP-binding state. Current paradigms of RAS signaling hold the PM localization of KRAS is constitutive and it is effectors that are drawn to RAS (e.g. translocation of RAF) not vice versa. Are the authors arguing that in this case the converse is true and that the localization of KRAS is driven by that of its effector?*

It has been difficult to discern specific co-localization between activated KRAS with AFDN in any polarized epithelial cell line we have worked with. There are several reasons for this: 1) both AFDN and KRAS are constitutively membrane localized in these lines independent of each other, as stated by the reviewer; 2) cells expressing activated KRAS detach from adjacent cells; and 3) live cell imaging has proven ineffectual for numerous reasons, primarily due to the unpredictable nature of KRAS-G12V expressing cells but also the requirement for high resolution z-planes and

the tendency of overexpressed AFDN to accumulate in the cytoplasm. To address the reviewers' points and validate that KRAS and AFDN co-localize in cells in a GTP-dependent manner we have taken several approaches. First, we have added numerous images of immunostained AFDN in KRAS-expressing MCF7 cells (the new **Supplementary Fig. 5c/d**). These images show that AFDN is no longer visible at cell-cell contacts in cells expressing KRAS-G12V, while it is retained at cell contacts in those expressing wild-type KRAS. The images demonstrate that cells expressing KRAS-G12V are no longer associated with adjacent cells, with empty space clearly observable in the surrounding space (i.e. detached). This addresses the reviewers first point above. Secondly, we reconsidered whether polarized epithelial cells with well-defined cell-cell contacts were the most suitable system to demonstrate co-localization between RAS GTPases and AFDN. As discussed above, HeLa cells lack discernable levels of endogenous AFDN, as well as apical-basal polarity and do not form adherens junctions. We exploited this to examine whether RAS could specifically recruit exogenous AFDN to the plasma membrane in a GTP-dependent manner. The new **Fig. 5c** shows how expression of activated KRAS, and not wild-type KRAS, co-localizes with AFDN at the plasma membrane. Indeed, most of the AFDN pool is recruited from the cytoplasm with a fraction remaining in the perinuclear region. We hope the reviewer agrees this is more robust demonstration that AFDN should be considered a RAS effector.

3) Fig. 5D. The altered localization of endogenous SCRIB as a consequence of silencing AFDN is described on p. 10 as "dispersed throughout the cytoplasm." But unlike the distribution of GFP that is homogeneous and clearly cytosolic, that of SCRIB is punctate consistent with a vesicular localization and should be described as such. Co-localization with Texas-red transferrin would allow an assessment as to whether these are endosomes. Caution must be taken in what is used for permeabilization of the cells (here 0.05% Tween-20) as this can alter the appearance of vesicles. It would be wise to also try 0.1% saponin.

We agree that the localization of SCRIB in AFDN KO MCF7 cells is more appropriately described as punctate and have changed the description in our revised manuscript. To assess whether the observed pattern is consistent with a localization to endosomes we expressed EGFP-tagged RAB5A, RAB7A or RAB11A in these cells followed by immunostaining for endogenous SCRIB. These report on early, late or recycling endosomes, respectively, and the new **Supplementary Fig. 6c** reveals that SCRIB does not co-localize with any of these markers. As noted above, others have observed a similar punctate pattern of endogenous SCRIB in different cell lines (and we see this in HeLa cells, which do not express significant levels of AFDN), but this experiment suggests these are not endosomes. Future delineation will hopefully shed light on where SCRIB is localized in the absence of AFDN.

4) Despite binding of several RAS family small GTPases, in their colocalization studies the authors restricted their analysis to KRAS4B. This is unfortunate. It would be informative to also study a RAS-related binding partner that is not normally localized exclusively to the PM. RAP2 fits the bill as the authors show strong binding to AFDN and this small GTPase has been localized to endomembrane (PMID: 1923507 and 19061864). It would also be interesting to determine if RHEB interacts with AFDN since this RAS family small GTPase is expressed on lysosomes.

To address this point we have studied co-localization of AFDN with the small GTPases RAP1B and RAP2C (the new **Fig. 5c**). We found that imaging these proteins in MCF7 cells came with many of the same caveats presented by AFDN-KRAS, and therefore used the HeLa cell line to determine if these protein partners co-localize (as described for KRAS above). RAP1B and RAP2C were the most prominent members of this family precipitated by the AFDN RA domains. Consistent with previous data, we observed RAP1B is distributed prominently throughout the cytoplasm with a fraction at the plasma membrane^{1,2}, while RAP2C is localized to the plasma membrane and on endomembranes^{3,4}. In both cases, we observed their complete co-localization with exogenous AFDN. Indeed, RAP2C-G12V stimulated recruitment of AFDN to the membrane in a manner very similar to that observed with KRAS-G12V.

1. Wilson, J. M., Prokop, J. W., Lorimer, E., Ntantie, E. & Williams, C. L. Differences in the Phosphorylation-Dependent Regulation of Prenylation of Rap1A and Rap1B. *Journal of Molecular Biology* 428, 4929–4945 (2016).
2. Ntantie, E. et al. An Adenosine-Mediated Signaling Pathway Suppresses Prenylation of the GTPase Rap1B and Promotes Cell Scattering. *Sci. Signal.* 6, (2013).
3. Duncan, E. D., Han, K.-J., Trout, M. A. & Prekeris, R. Ubiquitylation by Rab40b/Cul5 regulates Rap2 localization and activity during cell migration. *Journal of Cell Biology* 221, e202107114 (2022).
4. Meng, Z. et al. RAP2 mediates mechanoresponses of the Hippo pathway. *Nature* 560, 655–660 (2018).

Minor points:

5) *Fig. 2B. This figure sets up indirect immunofluorescence (iIF) staining of endogenous AFDN and SCRIB, which is used extensively throughout the paper. The specificity of antibodies used for iIF must always be validated by knockdown of the protein of interest, which is accomplished in Fig. 5 and this should be added to the legend of Fig. 2B.*

A statement was added to the figure legend.

6) *Fig. 4D. The authors write on p. 7 that with an n=5 they saw an 8-fold enhancement of SCRIB co-IP with AFDN upon expression of KRAS4B-G12V, but they do not report results for RAP1B or RAP2C, which appear to also induce some enhancement, albeit to a lesser extent (KRAS4B>>RAP2C> RAP1B). It would be informative to report the results for each of the interacting GTPases.*

We have added quantitation of SCRIB co-immunoprecipitation with AFDN in the new **Fig. 4g**. The plot shows SCRIB levels when co-expressed with KRAS-G12V, as well as RAP1B-G12V and RAP2C-G12V. Ratios are in comparison to SCRIB levels co-precipitated when AFDN is expressed with wild-type KRAS. While expression of KRAS-G12V results in an 8-fold increase in SCRIB association with AFDN, the levels are indeed lower for RAP2C (2.6-fold increase) and RAP1B (1.7-fold increase).

7) *Fig. S6B. It is very difficult to see the AFDN staining.*

We have increased intensity of the AFDN signal in these images (now **Supplementary Fig. 7b**).

8) *Fig. 6C,D. These z projections are convincing that true colocalization of AFDN and SCRIB requires PDZ1 and FHA, but would be even more so if they were subjected to analysis with Pearson's coefficient.*

We have calculated Pearson's coefficients to quantify the levels of endogenous SCRIB co-localized with EGFP-tagged wild-type AFDN or the Δ FHA variant (in AFDN KO cells), or endogenous AFDN with EGFP-tagged wild-type SCRIB or its Δ PDZ1 variant (in SCRIB KO cells). These calculations were performed on $n \geq 8$ images similar to those presented in **Fig. 7c/d**. The results are presented in the new **Supplementary Fig. 7c**. This quantitation resolved a significant difference in the co-localization of endogenous SCRIB/AFDN with the respective wild-type rescues compared with domain deletions. The analysis has more noise in the AFDN KO cells due to the punctate nature of SCRIB localization, but there is clearly a deficiency in SCRIB co-localization with the AFDN Δ FHA rescue compared to wild-type.

9) *The authors refer to KRAS throughout but they mean KRAS4B. They do not study KRAS4A. Since these splice variants differ only in their HVRs that direct subcellular trafficking this should be acknowledged.*

We have acknowledged this in the Results section (page 5) and in the Methods section (*Plasmid Constructs and Antibodies*).

10) *To be a true effector of a small GTPase, three conditions must be met. The effector must bind directly to the GTPase, the binding must depend on GTP-loading of the GTPase, and the binding must in some way change the conformation or activity of the effector. RAF and HK1 meet all of these criteria but the third has been lacking for AFDN. In Fig. 4D the authors establish for the first time a change in the properties of AFDN induced by KRAS, confirming that AFDN is a bone fide effector. This should be discussed.*

The point is well taken, and we have reinforced this view in the Discussion (bottom of the first paragraph). It is still not completely clear how RAS activates most effectors, including RAF from a completely mechanistic perspective, though cryo-EM and modelling data are making significant progress with this. It does appear that GTPase binding to AFDN (particularly KRAS) increases its complex with SCRIB, and while only a structure of the full proteins complexed with RAS will reveal the mechanism behind this 'activation', it does appear to satisfy the third condition for being a true effector.

11) *Because AFDN is unique in possessing tandem RA domains that, in principal, could bind two GTPases the authors have a unique opportunity to ask if either or both are required for the effect of KRAS seen in Fig. 4D. Indeed, it would be interesting to determine if RAS binds to one and RAP2 to the other RA domain.*

To address this point we generated bacterial expression constructs to allow purification of AFDN RA1 or RA2 alone. We reveal the specificity of the two individual RA domains for RAS and RAP GTPases, in contrast to the RA1/2 protein, in the new **Fig. 4d**. The first RA domain of AFDN demonstrated a very similar binding profile to the dual RA1/2 construct, but the levels of GTPase precipitated were considerably lower than with the tandem domains, KRAS in particular. The RA2 binding profile is more restricted, with only KRAS, RAP2B and RAP2C showing significant interaction. Overall, the results suggest that the tandem domains do provide an avidity for most of these GTPases which drives tighter binding than with either domain in isolation. RAP1B seems an exception, whereby binding is generated predominantly through RA1. These data are now described in the Results section.

12) The change in kinetics of ERK and AKT activation downstream of EGF signaling upon silencing AFDN or SCRIB shown in Fig. 7D,E is interesting but the authors do not comment on possible mechanisms. Interestingly they parallel the differential effects of NGF versus EGF in PC12 cells where only the former induces sustained ERK activation (PMID 7834738).

Indeed, the duration of ERK activation has long been recognized as a foremost feature of signalling through the MAPK pathway, and potentially as a determinant of proliferation vs differentiation outcomes. Interestingly, the sustained activation of MAPK signalling induced by NGF (pointed out by the reviewer) is ostensibly dependent on RAP1¹, making the AFDN association with RAS and RAP of particular relevance to this phenomenon. As this will be a focus of future work and we do not currently have clear mechanistic data elucidating how loss of AFDN disrupts ERK activation, we have added only a short speculation to the Discussion section (paragraph 4). It seems clear that AFDN association with RAS at the plasma membrane will compete with binding of other effectors to activated RAS, including the RAF kinases, and this is the most likely explanation for the observed signalling defect. It is also probable that competition between activated RAS and RAP GTPases for the AFDN RA domains will further impact effector binding to the individual GTPases themselves. For SCRIB, it is more difficult to reason how loss of expression induces nearly the identical defect as loss of AFDN. It is possible that the RAS-AFDN-SCRIB module is more stable than the RAS-AFDN module alone, and our data supports this. Thus, SCRIB could be an important determinant of competition between AFDN and RAF effectors for activated RAS.

1. York, R. D. et al. Rap1 mediates sustained MAP kinase activation induced by nerve growth factor. *Nature* 392, 622–626 (1998).

REVIEWERS' COMMENTS

Reviewer #1 (Remarks to the Author):

The authors significantly approved the quality of their manuscript.

Reviewer #4 (Remarks to the Author):

The authors have addressed my comments and those of three other reviewers with new data, a revised manuscript and a well-argued rebuttal. They have done an outstanding job and the revised manuscript is significantly improved. This interesting, comprehensive, and significant paper is suitable for publication in Nature Communications.

I have two minor comments not intended to diminish the outstanding work presented but for future analyses should the authors carry this work forward. The authors write in their rebuttal: "To satisfy the reviewers request, we coexpressed EGFP-AFDN and Cherry-SCRIB in these cells (the new Supplementary Fig. 1d), or appropriate controls, to avoid fixation/permeabilization effects." However, the authors did not eliminate fixation/permeabilization effects because rather than image these cells alive, as would have been preferable, they fixed the cells, apparently to allow Hoechst staining. But nuclear localization with Hoechst adds little if anything to the analysis and would be far outweighed by the power of live cell imaging.

One other caveat along these lines, the authors refer to the localizations of proteins like GFP alone and GFP-AFADIN in cells not expressing oncogenic KRAS as "cytoplasmic." But the cytoplasm includes both cytosol and all membrane bound organelles (endomembrane). Live cell imaging allows one to clearly visualize the cytosol (organelles appear to be negatively imaged), which is an important descriptive term for molecules like AFADIN that translocate not from endomembrane to plasma membrane but rather from cytosol to plasma membrane.